



# Sea Level Dynamics and Coastal Erosion in the Baltic Sea Region

Ralf Weisse[1], Inga Dailidiene[2], Birgit Hünicke[1], Kimmo Kahma[3], Kristine Madsen[4], Anders Omstedt[5], Kevin Parnell[6], Tilo Schöne[7], Tarmo Soomere[6], Wenyan Zhang[1], Eduardo Zorita[1]

[1]Helmholtz Zentrum Geesthacht, Centre for Materials and Coastal Research, Geesthacht, 21502, Germany
[2]Klaipeda University, Faculty of Marine Technology and Natural Sciences, Klaipeda, LT-92294, Lithuania
[3]Finnish Meteorological Institute, Helsinki, Finland
[4]Danish Meteorological Institute, Copenhagen, 2100, Denmark
[5]University of Gothenburg, Department of Marine Sciences, Box 460, SE-405 30 Göteborg, Sweden
[6]Tallinn University of Technology, School of Science, Department of Cybernetics, Tallinn, 12618, Estonia
[7]German Research Centre for Geosciences GFZ, Potsdam, 14473, Germany

*Correspondence to*: Ralf Weisse (ralf.weisse@hzg.de)

**Abstract.** There are a large number of geophysical processes affecting sea level dynamics and coastal erosion in the Baltic Sea region. These processes operate on a large range of spatial and temporal scales and are observed in many other coastal regions worldwide. Together with the outstanding number of long data records, this makes the Baltic Sea a unique laboratory for advancing our knowledge on interactions between processes steering sea level and erosion in a climate change context. Processes contributing to sea level dynamics and coastal erosion in the Baltic Sea include the still ongoing visco-elastic response of the Earth to the last deglaciation, contributions from global and North Atlantic mean sea level changes, or from wind waves affecting erosion and sediment transport along the subsiding southern Baltic Sea coast. Other examples are storm surges, seiches, or meteotsunamis contributing primarily to sea level extremes. All such processes have undergone considerable variations and changes in the past. For example, over the past about 50 years, the Baltic absolute (geocentric) mean sea level rose at a rate slightly larger than the global average. In the northern parts, due to vertical land movements, relative sea level decreased. Sea level extremes are strongly linked to variability and changes in the large-scale atmospheric circulation. Patterns and mechanisms contributing to erosion and accretion strongly depend on hydrodynamic conditions and their variability. For large parts of the sedimentary shores of the Baltic Sea, the wave climate and the angle at which the waves approach the nearshore are the dominant factors, and coastline changes are highly sensitive to even small variations in these driving forces. Consequently, processes contributing to Baltic sea level dynamics and coastline change are expected to vary and to change in the future leaving their imprint on future Baltic sea level and coastline change and variability. Because of the large number of contributing processes, their relevance for understanding global figures, and the outstanding data availability, we argue that global sea level research and research on coastline changes may greatly benefit from research undertaken in the Baltic Sea.



## 1 Introduction

Regional climate change in the Baltic Sea basin has been systematically assessed in two comprehensive assessment reports: BACC I (BACC Author Team, 2008) and BACC II (BACC II Author Team, 2015) initiated by BALTEX and its successor Baltic Earth (https://baltic.earth). As a follow-up, the present study represents one of the thematic Baltic Earth Assessment

Reports (BEARs) which consists of a series of review papers summarizing and updating the knowledge around the major Baltic Earth science topics. Being part of the series, this study concentrates on sea level dynamics and coastal erosion in the Baltic Sea region.

The Baltic Sea is an intra-continental, semi-enclosed sea in northern Europe that is connected to the Atlantic Ocean only via the narrow and shallow Danish Straits. With an area of about 393,000 km$^2$ and a volume of about 21,200 km$^3$

(Leppäranta and Myrberg, 2009), it contributes less than a tenth of a percent to the area and the volume of the global ocean (Eakins and Sharman, 2010).

While from a global perspective sea level and coastline changes in the Baltic Sea may appear irrelevant, they have received considerable attention over the last centuries and decades for several reasons:

1.  Historically, changes in sea levels and coastlines have influenced Baltic Sea harbors, settlements, and
economic activity over millennia. As a result, the area comprises not only some of the longest available tide-gauge records worldwide, but also much longer observational evidence that challenged our understanding of sea level dynamics and land movements associated with the glacial isostatic adjustment (GIA) (e.g., BIFROST project members, 1996), and contributed significantly to our present understanding of large scale sea level changes on a global scale.

2.  Processes and forcing contributing to Baltic sea level dynamics and coastline change substantially vary over short distances (Harff et al., 2017). Time scales of processes and forcing vary considerably, ranging from a few seconds to millennia. Again, this enables researchers to study a wide range of phenomena with larger and global relevance.

3.  Finally, and from a regional perspective, regional mean and extreme sea level changes and erosion represent
important indicators of regional climate variability and change. Any long-term change in mean or extreme sea levels as well as in erosion and accretion will have an immediate impact on society, influencing sectors such as coastal protection, shipping, or development of offshore renewable energy resources among others (e.g., Weisse et al., 2015).

Historically, considerable progress in sea level research worldwide was made based on early Baltic Sea observations

that indicated that sea levels were falling. Stones carved with runic texts, linking them to the coast, were found quite far away from the present-day coastline. Shallow harbors were gradually abandoned as the water level apparently declined. In the 18$^{th}$ century, Celsius (1743) estimated the rate of falling water levels based on so-called seal rocks (Figure 1,





left). Seal rocks were economically important for seal hunting and are therefore well described in the written records (Ekman, 2016).

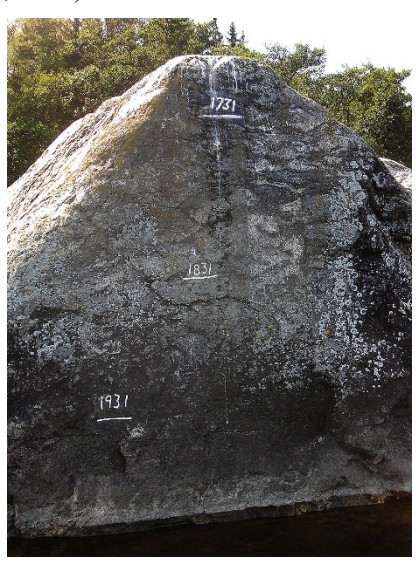

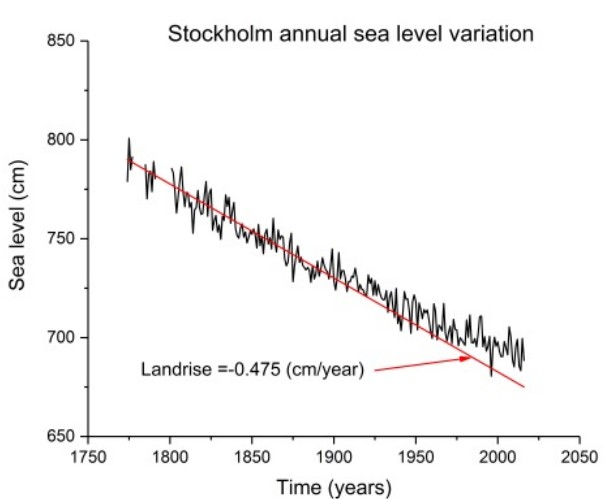

**Figure 1: (Left) The Celsius seal rock at Lövgrunden outside the Swedish city of Gävle on the Bothnian Sea coast (Ekman, 2016). The water level today is about 2 meters below the 1731 mark (photo courtesy of Martin Ekman). (Right) Stockholm annual sea level variations (black) and land rise (red) according to Ekman (2003) [redrawn from Omstedt (2015)].**

The reason for the sinking water levels was unclear and debated until it was understood, that in the past thick layers of ice had covered Scandinavia and that sea levels were not falling but instead the land was rising elastically after the

ice cover disappeared. The idea of postglacial uplift was proposed in the mid-19th century by Jamieson (1865) and then later by others, although the causes of the uplift were strongly debated. Another major progress in ideas was not possible until new knowledge of the thermal history of the Earth due to changes in Sun-Earth orbital motions was available in the late 19$^{th}$ century and early 20$^{th}$ century (Milanković, 1920).

Figure 1 (right) shows observed Baltic sea level change and land rise for Stockholm, one of the longest available tide-

gauge records comprising almost 250 years of data. Apart from the long-term trend, substantial variability on different time scales is inferred. For the Baltic Sea, processes contributing to such variability can be separated into processes that alter the volume of the Baltic Sea and/or the total amount of water in the basin, and processes that redistribute water within the Baltic Sea (Samuelsson and Stigebrandt, 1996). From analyses of tide-gauge data and dynamical consideration we know, that processes with characteristic time scales of about half a month or longer can

change the volume of the water in the Baltic Sea. Due to the limited transport capacity across the Danish Straits, processes with shorter time scales primarily redistribute water within the Baltic Sea (Johansson, 2014; Soomere et al., 2015; Männikus et al., 2019). At longer time scales, North Atlantic mean sea level changes and effects from large-scale atmospheric variability have the strongest influence on Baltic mean sea level variability and change apart from changes caused by movements in the Earth's crust due to the GIA. About 75% of the basin-average mean sea level





change externally enters the Baltic Sea as a mass signal from the adjacent North Sea (Gräwe et al., 2019). On shorter
time scales, atmospheric factors such as precipitation and wind are the primary drivers of sea level variations
occurring within the Baltic Sea (Hünicke and Zorita, 2006). Figure 2 provides a sketch of processes relevant to Baltic
sea level dynamics and coastline changes.

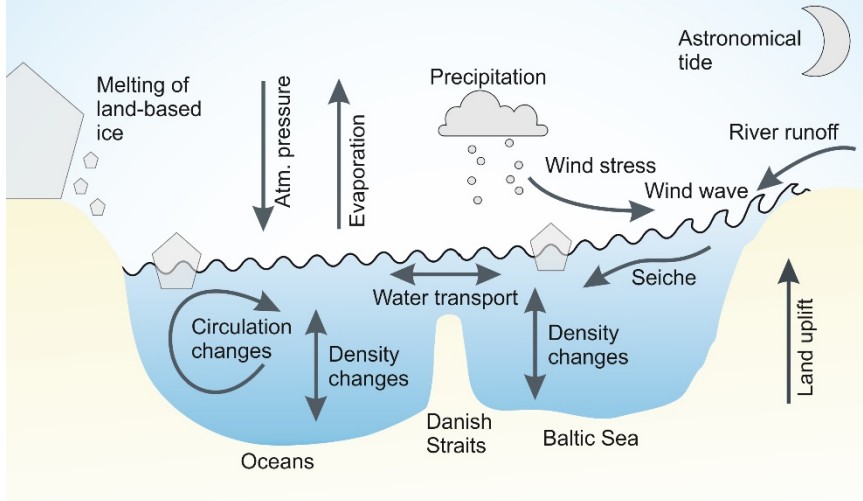

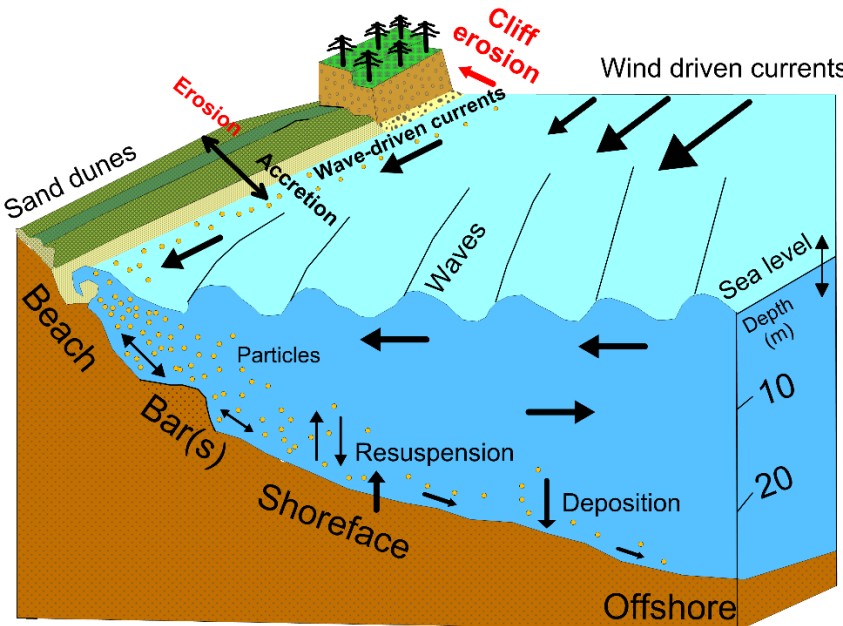


**Figure 2: Processes contributing to sea level variability and change in the Baltic Sea [top, redrawn and modified from Johansson
(2014)] and main coastal topographic features and erosion/accretion processes at the Baltic Sea shores [bottom, redrawn and
modified from Harff et al. (2017)].**

The primary control of the position of the shoreline is sea level (Harff et al., 2017). Much of the Baltic Sea shoreline

(particularly in Finland and Sweden) is rock or consolidated sediments, which typically change over decades to



millennia. Shorelines composed of unconsolidated sediments typically erode when sea level rises and accrete when sea level falls. However, several other factors modify the control that sea level exerts. Sediment availability and supply are fundamental. A useful conceptual model is the coastal sediment budget, with sediment transport being considered within sediment compartments, which operate essentially independently, particularly with respect to

alongshore sediment transport. Sediment compartments may be very small (hundreds of meters of shoreline) but can range up to hundreds of kilometers. Where the sediment budget is in deficit, erosion occurs; where it is in surplus, the shoreline accretes. Overlying any long-term trend are shorter-term changes to the shoreline. Changes that occur after a storm event are typically called erosion, although technically if the sediment remains in the active sediment compartment, this use of the term in this context may be seen as inaccurate. The Baltic Sea differs from many other

coastal locations. In particular, short wind fetches mean that swell waves are generally insignificant (Broman et al., 2006; Soomere et al., 2012) and waves often approach the shoreline at a large angle (Soomere and Viška, 2014). That means that the Baltic Sea shorelines are very sensitive to the wind direction. Wave periods are short, leading to the situation where waves can approach the shore at high angles, and the effects of an individual storm on beach processes and sediment transport within a compartment can vary considerably depending on wind (and therefore

wave) characteristics, overlying the effects of short term elevated water levels.

Being part of the thematic Baltic Earth Assessment reports, this study concentrates and reviews knowledge on sea level dynamics and coastal erosion in the Baltic Sea region. Section 2 provides an overview of the current state of knowledge for both, mean and extreme sea levels as well as for coastal erosion and sedimentation. This includes brief recaps of sources of data and relevant processes along with descriptions of variability and past, ongoing, and

potential future changes. Section 3 addresses several knowledge gaps, that emerged from reviewing the available publications. Eventually, conclusions and key messages are presented in Section 4.

## 2 Current state of knowledge

### 2.1 Mean sea level

### 2.1.1 Sources of data

The primary sources for measuring mean sea level changes are tide gauges and radar altimetry from satellites. Both measure two different quantities, namely *relative* and *absolute (geocentric)* sea levels. Absolute sea level refers to the height of the sea surface relative to a geocentric reference such as the reference ellipsoid and is derived from satellite altimetry or global navigation satellite system (GNSS)-controlled tide gauges (Schöne et al., 2009; Schöne et al., 2011). Relative sea level refers to the height of the sea surface relative to the seafloor (or a local benchmark on land) and is derived from tide gauges or sea

level reconstructions. Contrary to absolute sea level, relative sea level is thus influenced both by variations in the height of the sea surface and by land uplift or subsidence.



The tide gauge network of the Baltic Sea is one of the most densely spaced and longest-running networks in the world. Many stations have been in continuous operation since the early 19[th] century and some stations provide monthly averages for over 200 years (Ekman, 2009; Bogdanov et al., 2000; Kowalewska-Kalkowska and Marks, 2011). There are about 45 operational
tide gauges with more than 60 years of data and with very good coverage of the western, southwestern, and northern Baltic Sea coasts (Hünicke et al., 2015). Eight additional gauges with data since 1961 in Latvian waters were recently added to this data set (Männikus et al., 2019). Monthly averaged data is often provided to, and open access through, the Permanent Service for Mean Sea Level (PSMSL; e.g., Holgate et al., 2013). The Copernicus Marine Environment Monitoring Service collects data in hourly resolution from a range of national stations in hourly resolution with daily updates, but with varying
quality control. The EMODnet Physics portal provides access to both data sources, and thus combines a substantial amount of long-term monthly and higher frequency Baltic sea level measurements. The tide gauges measure relative to specific reference levels that differ between Baltic countries for historical reasons (Ekman, 2009). Additionally, uncertainties in local reference levels are introduced by, e.g. different measuring techniques and sampling frequencies (Ekman, 2009), changes in the reference points due to relocation of benchmarks, or man-made changes of the tide-gauge surroundings, such as coastal
or port development (e.g., Bogdanov et al., 2000). Phenomena occurring on timescales of years to decades, such as subsidence, for example, the sinking of piers due to unstable foundations, or land sinking due to groundwater or gas extractions, can be other sources of uncertainties (Hünicke et al., 2015). Subsidence can further result from processes such as compaction of unconsolidated alluvial soils, oxidation of organic materials, or increased surface load due to construction activities (e.g. Pelling and Blackburn, 2013).

Radar satellite altimetry from different satellite missions is available continuously since 1991 (e.g., Schöne et al., 2010). Presently, the longest and most sound dataset is derived from a combination of the consecutive dual-frequency missions Topex/Poseidon, Jason-1, -2, and -3 (Legeais et al., 2018). The orbit repeats itself every 10 days allowing the construction of continuous time series since 1993. The missions form a pattern of ascending and descending orbits with an average cross-track distance of 100 km (at 60°N). Starting in 1991 with the European mission ERS-1 (followed by ERS-2, and the dual-
frequency mission ENVISAT (ESA) and AltiKa (India/France)) the Baltic Sea is also mapped every 35 days with an average cross-track distance of 55 km (at 60°N). With the launch of ESA's CryoSat-2 in 2010, a new altimetry concept was established using a synthetic aperture interferometric radar altimeter (SAR/SARin) to especially map ice parameters but also sea surface heights closer to the coastlines. In 2016 the European COPERNICUS program launched a mission on Sentinel-3A (followed in 2018 by Sentinel-3B) carrying forward the uninterrupted altimetry program with 27 day repeat periods. The
recent launch of Sentinel-6A Michael Freilich in November 2020 will continue the important missions with unprecedented accuracy.



### 2.1.2 Variability, change, and acceleration of Baltic Sea mean sea level

Long-term changes in the Baltic Sea mean sea level are the result of several processes including thermo- and halosteric effects, long-term changes in wind and surface air pressure, ocean currents, variations in freshwater input, and gravitational

effects. Effects may arise from contributions outside and inside the Baltic Sea (Figure 2).

Global mean sea level rise since the beginning of the last century was estimated from tide gauge records with rates ranging between about 1–2 mm yr$^{-1}$ (e.g., Oppenheimer et al., 2019). For the era of continuously operated satellite radar altimeters beginning in 1991, higher estimates ranging between about 3–4 mm yr$^{-1}$ are reported (e.g., Nerem et al., 2018; Oppenheimer et al., 2019). These trends cannot directly be compared because of the strong decadal variability inherent in the records (e.g.,

Albrecht et al., 2011) and their different spatial representativeness: While tide gauge data are more representative for coastal changes, satellite data typically reflect open ocean changes and variability. Also, inhomogeneous data, comprising a few tide-gauge records in the 19[th] and early 20[th] centuries and satellite data with nearly global coverage in the 21[st] century, hamper quantification and comparability of trends (Jevrejeva et al., 2008).

For the Baltic Sea, an increase in absolute mean sea level of about 3.3 mm yr$^{-1}$ was estimated based on available multi-

mission 1992–2012 satellite altimetry data (Stramska and Chudziak, 2013). This figure is broadly consistent with the global average rate over that period. Current (1993-2015/2017) altimetry derived Baltic Sea mean sea level trends are still comparable with the global average, with a need for dedicated altimetry processing in the coastal region (Madsen et al., 2019a; Table 1).

Relative mean sea level changes and their secular trends are strongly affected by vertical land movements and vary

considerably across the Baltic Sea. Geologically, the Baltic Sea region is divided into the uplifting Fennoscandian Shield in the North and the subsiding lowland parts in the South (Harff et al., 2007). Land uplift rates in the northern parts are in the order of several millimeters per year and are comparable to the climatically induced rates of sea level rise in the 21[st] century. Relative mean sea level trends in the Baltic Sea show a corresponding north-south gradient, reflecting these crustal deformation rates due to GIA: In the northern parts, relative mean sea level decreases with a maximum rate of about 8.2 mm

yr$^{-1}$ in the Gulf of Bothnia (Hünicke et al., 2015). This corresponds to the area with maximum GIA-induced crustal uplift (Peltier, 2004; Lidberg et al., 2010). In the southern Baltic Sea, relative mean sea level increases at a rate of about 1 mm yr$^{-1}$ with a gradient in a northeasterly direction (Richter et al., 2012; Groh et al., 2017). These findings are supported by numerous studies analyzing Baltic relative mean sea level trends on a national basis (Suursaar et al., 2006; Dailidienė et al., 2012; Männikus et al., 2019). Different observation periods and analysis techniques hamper comparison to some extent. A

more comprehensive figure on Baltic sea level trends and variability was provided by Madsen et al. (2019a) who took a statistical approach to combine data from century-long tide gauge records with results from hydrodynamic modeling based on atmospheric reanalysis data, and compared the results with those from satellite altimetry records (Table 1).





**Table 1: Rates of mean sea level rise for the Baltic Sea and estimated uncertainty (one standard deviation range) based on (Madsen et al., 2019a).**

|  | Mean sea level rise (mm/year) | Uncertainty (mm/year) |
|---|---|---|
| Statistical model, 1900-1999 | 1.3 | 0.3 |
| Statistical model, 1915-2014 | 1.6 | 0.3 |
| Statistical model, 1993-2014 | 3.4 | 0.7 |
| Satellite data (CCI[1]), 1993-2015 | 4.0 | 1.3 |

[1] Here CCI refers to ESA Sea Level CCI ECV v2.0 (Quartly et al., 2017; Legeais et al., 2018)

Atmospheric forcing in the form of wind and precipitation may affect the basin average sea level through changes in the total volume of water, but also the internal distribution of water volume within the Baltic Sea basin. Winds, and more particularly the strength of the westerly winds, modulate the exchange of water masses with the North Sea (Gräwe et al., 2019). This is due to the alignment of the prevailing wind directions in this region with the geographical orientation of the connecting straits between the Baltic and the North Seas: Stronger than normal westerly winds push water masses into the Baltic Sea, raising overall sea level. Precipitation anomalies may, through the corresponding changes in salinity and water density, also affect the distribution of the Baltic mean sea level (Hünicke and Zorita, 2006). On average, there is a strong salinity increase from the Northeast to the Southwest of the Baltic Sea. This average salinity gradient, together with the volume of the freshwater input and the effects of the prevailing westerly winds, gives rise to a corresponding sea level variation of about 35–50 cm across the Baltic Sea (Ekman and Mäkinen, 1996). Any changes in the hydrological cycle, either in the long term or in the form of episodes associated with strongly increased or decreased river runoff or precipitation/evaporation over the Baltic Sea, have the potential to substantially modify this salinity gradient (Kniebusch et al., 2019) and thus the distribution of sea level anomalies and their variability in space and time.

The relation between Baltic sea level variability with the state of the large-scale atmospheric circulation has been the subject of numerous studies (Kahma, 1999; Johansson et al., 2001; Lehmann et al., 2002; Dailidiene et al., 2006; Hünicke and Zorita, 2006; Suursaar et al., 2006; Johansson and Kahma, 2016; Chen and Omstedt, 2005, 2005; Omstedt et al., 2004). Mostly, these studies focused on relations between the NAO[1] and Baltic sea level. Generally, correlations strongly vary by region with higher values in the northern and the eastern parts and smaller values in the southern parts of the Baltic Sea. Moreover, correlations were found to be rather variable over the 20th century, including periods with both high and very low values. These results initiated a search for atmospheric patterns that might be connected better, and be more stable related to sea level in the Baltic Sea. Karabil et al. (2018) for example, suggested such a pattern in which a gradient in sea level pressure anomalies is oriented from southwest to northeast directions and in which the two centers of action are located over the Baltic Sea and the Bay of Biscay. Compared to the NAO, the correlation of this pattern with the Baltic Sea mean sea

[1]   The *North Atlantic Oscillation* (NAO) basically describes a meridional pattern in sea level pressure (SLP) with higher than normal  SLP around the Azores and lower than normal SLPs over Island and vice versa. Variability of this pattern is physically linked to the intensity of the westerlies in the European region.



level was found to be more stable, suggesting that this pattern is better suited to describe Baltic sea level variability associated with large-scale atmospheric changes. Karabil et al. (2018) suggested that the possible mechanism behind this link is not so much the direct effect of wind on the ocean surface, but is rather associated with the inverse barometer effect.

The contribution of the different mechanisms behind current sea-level trends can be also estimated from the analysis of simulations with regional ocean models driven by observed atmospheric and global mean sea level forcing (Gräwe et al.,
2019). This approach has the advantage that the effects of land-movement on the Baltic sea level are explicitly neglected so that the contribution from other factors can be disentangled. In the simulation of Gräwe et al. (2019), the Baltic sea level rose at a rate of about 2 mm yr$^{-1}$ over the past 50 years, a rate that is slightly larger than the global average. Most of this sea level rise in the Baltic Sea was caused by a corresponding increase of sea level in the North Atlantic Ocean. Model results along with data from Latvian waters (Männikus et al., 2020) further suggest a heterogeneous pattern of sea level rise, with larger
rates in the northern and smaller rates in the southeastern Baltic Sea that appears to be the result of a poleward shift of atmospheric pressure systems (Gräwe et al., 2019).

For the future, global mean sea level rise is expected to have the largest impact on future Baltic sea level changes (Grinsted, 2015; Hieronymus and Kalén, 2020). It is expected, that most of the future Baltic absolute sea level rise will be strongly linked with corresponding large-scale changes in the North Atlantic and the factors modulating these
changes. These factors are mainly the thermal expansion of the water column, contributions from melting of the Antarctic ice-sheet (Grinsted, 2015), and imprints from the variability and change of the Atlantic Meridional Overturning Circulation  (Börgel et al., 2018). More recent estimates of potential contributions from Antarctic melting point to slightly higher sea level rise in the Baltic Sea when compared to the global average (Hieronymus and Kalén, 2020). However, this is somewhat compensated by lower recent estimates of dynamical sea-level rise in the North Atlantic region
(Hieronymus and Kalén, 2020). Stronger winds and increased run-off may further contribute to future Baltic sea level rise in the order of some cm (Meier, 2006; Hünicke, 2010; Johansson, 2014; Karabil, 2017; Pellikka et al., 2018). Future changes in the Baltic Sea freshwater budget may further affect the regional distribution of Baltic mean sea level (Hünicke et al., 2017). If present links between atmospheric forcing and Baltic mean sea level are extrapolated into the future, their contribution to future Baltic mean sea level can be estimated from climate projections. Presently
such contributions are thought to be minor (Weisse and Hünicke, 2019). As GIA continues and will persist to dominate relative mean sea level changes in the northern parts of the Baltic Sea, these areas are expected to see a continued, although decelerated, decrease in relative mean sea level in the future, while the southern parts are expected to experience a relative mean sea level rise slightly exceeding the global average (Räisänen, 2017).

One of the expected implications of anthropogenic climate change would be an acceleration in sea level rise over
time. So far, global mean sea level rise seems to have accelerated over the 20th century (e.g., Nerem et al., 2018; Oppenheimer et al., 2019) although there is still ongoing debate (Kleinherenbrink et al., 2019; Veng and Andersen, 2020). Robust detection of acceleration is to some extent hampered by changes in the measurement system: While most of the data over much of the 20th century originate from coastal tide gauges mostly in the Northern Hemisphere,



since 1991 satellite altimetry provides a nearly global figure mostly from the open ocean. Differences in estimates of
satellite intra-mission biases also have substantial effects on estimated acceleration rates (Kleinherenbrink et al.,
2019). As tide gauges in the Baltic Sea provide some of the longest, best quality controlled, and more homogeneous
records, they may provide some support in assessing acceleration. From a statistical analysis of long Baltic sea level
records, Hünicke and Zorita (2016) could indeed detect an acceleration in sea level rise. While the tests were not
powerful enough to detect a significant acceleration in most of the records taken individually, results are robust for
the set of gauges against different definitions of acceleration. The overall magnitude of the acceleration is small, and
if continued unchanged over the whole 21$^{st}$ century would add just a few centimeters to the sea level rise resulting
from a constant rate.

Acceleration of Baltic sea level rise also displays a spatial pattern with accelerations in the northeastern parts three
times as large as those in the southwest. The spatial structure is compatible with what would be expected from an
expected deceleration of GIA. However, applying the theoretical deceleration calculated with models of the dynamics
of the Earth's crust (Spada et al., 2014) the resulting order of magnitude is still too small to explain the apparent
acceleration of relative coastal sea level. It has to be kept in mind that most tide-gauges are affected by the rather
strong noise of local vertical movements, and thus the detection of a second-order change, like acceleration, is
challenging.

## 2.2 Extreme sea levels

### 2.2.1 Sources of data

Sea level extremes are measured using the same primary devices as for mean sea level; that is, tide gauges and radar
altimetry. For wind waves, also data from visual observations, wave buoys, or radars mounted on platforms are common.
Spatial coverage of instrumental wave measurements in the Baltic Sea is scarce and limited to a few regions. So far, no long-
term data exists from the central and eastern Baltic Proper and the Gulf of Riga (Suursaar et al., 2012). In the northern parts,
measurements are often limited to ice-free periods as buoys are usually removed during winter to prevent damage from ice.
Homogeneity of data may be problematic due to effects from the relocation of measurement sites or the replacement of
instruments (Hünicke et al., 2015). Wave data from measurements are often used in combination with reconstructions from
numerical wave hindcasts (Cieślikiewicz and Paplińska-Swerpel, 2008; Nikolkina et al., 2014; Björkqvist et al., 2018;
Soomere et al., 2012), where, in combination, they provide a reasonable characterization of open sea wave fields.

### 2.2.2 Variability and change of Baltic sea level extremes

Baltic sea level extremes occur over a wide range of spatial and time scales. Contributions may arise from wind wave
run-ups occurring at a local scale with variations in the range of seconds up to variations in the volume of the entire
Baltic Sea with formation time scales of up to a few weeks and even longer persistence (Soomere and Pindsoo, 2016).



They are generated mostly by meteorological and to some extent by astronomical factors (Weisse and Hünicke, 2019). From a climate perspective, this indicates that any relevant change in meteorological forcing may be associated with corresponding changes in Baltic sea level extremes.

The most prominent and most relevant phenomena contributing to sea level extremes in the Baltic Sea are storm surges, wind waves, and a "preconditioning" that leads to increased water volumes in the entire Baltic Sea. This

preconditioning is associated with periods of prevailing westerly winds that increase the sea level gradient across the Danish Straits. In turn, the increased sea level gradient leads to higher inflow and higher Baltic Sea water volumes (Samuelsson and Stigebrandt, 1996). Flows across the Danish Straits can reach values of up to about 25 km$^3$ day$^{-1}$ in both directions (Leppäranta and Myrberg, 2009), which corresponds to a sea level change of about 6 cm day$^{-1}$ over the entire Baltic Sea (Johansson, 2014). Major inflow events are associated with typical volumes in the order of about

100 km$^3$, corresponding to a Baltic sea level increase of about 24 cm (Matthäus and Franck, 1992). Typically, such variations have time scales of about 10 days and longer (Soomere and Pindsoo, 2016) while atmospheric variability on shorter time scales primarily leads to a redistribution of water masses within the Baltic Sea basin (Kulikov et al., 2015) or between the Baltic Proper and the Gulf of Riga (Männikus et al., 2019).

Storm surges represent a substantial threat for the low-lying coastal areas of the Baltic Sea, in particular in the

southwestern parts (Wolski et al., 2014), the Gulf of Finland (Suursaar and Sooäär, 2016; Averkiev and Klevannyy, 2010), the Gulf of Riga (Suursaar and Sooäär, 2016; Männikus et al., 2019), and the Gulf of Bothnia (Averkiev and Klevannyy, 2010). They are primarily caused by strong onshore winds during storms and secondarily by the action of spatially varying atmospheric pressure on the sea surface. They may last from several hours to almost a day. Because of the seasonal cycle in wind speed, storm surges are highest and most frequent during fall and winter (Weidemann,

2014). The presence of sea ice may substantially reduce the effectiveness of wind in generating storm surges. In winter in the Gulf of Bothnia, the piling-up of water is strongly suppressed by the existence of sea ice (Zhang and Leppäranta, 1995). When preconditioning exists, that is high Baltic Sea water volumes, even moderate wind and wind surges may lead to coastal sea level extremes (e.g., Weisse and Weidemann, 2017).

Wind waves in the Baltic Sea show a pronounced seasonal cycle, with higher values in winter and smaller values in

summer. This behavior is associated with a corresponding seasonal cycle in wind speed (Björkqvist et al., 2018) and wave energy flux (Soomere and Eelsalu, 2014). Generally, waves are higher in open waters than in coastal waters. To date, the highest measured waves in the Baltic Sea were reported from a wave buoy in the northern Baltic Proper. During a storm in December 2004, this buoy recorded a significant wave height of 8.2 m (Björkqvist et al., 2018; Tuomi et al., 2011). During windstorm Gudrun in 2005, measured significant wave heights reached 7.2 m in the

Baltic Proper and 4.5 m in the Gulf of Finland. For this storm and away from sensors, model simulations suggested the existence of extreme significant wave heights of up to 9.5 m off the northwestern coast of Latvia (Soomere et al., 2008). More recently, very high waves with significant wave heights up to 8.1 m were also recorded in the Sea of



Bothnia[2]. However, in the Bothnian and the northern Baltic Seas, the seasonal presence of sea ice typically modifies the wave climate and limits wave heights during the freezing season (Tuomi et al., 2011).

In the coastal zone, the height of the wave extremes is considerably smaller, but other wave-related processes such as wave set-up or run-up (swash) can make substantial contributions to short-term sea level extremes. Wave set-up refers to a wave-induced increase in the mean water level caused by the release of momentum from the dissipation of a long sequence of breaking waves and swash is caused by the run-up of single waves on the beach (Melet et al., 2018). For example, for some coasts in the Gulf of Finland (Soomere et al., 2013; Soomere et al., 2020) and along the

shores of the western Estonian archipelago (Eelsalu et al., 2014) potential wave set-up may reach values of up to 70–80 cm. Moreover, the shape of the shoreline and irregular bottom topography may strongly modify local wave conditions (Tuomi et al., 2012; Tuomi et al., 2014), and wave set-up and run-up may both have a strong influence on the level of the flooding and erosion at the shore (Dean and Bender, 2006). Coastal wave climate, its impacts, and its long-term changes thus strongly depend on location.

Changes in the Baltic Sea volume, storm surges, and extreme sea levels are tightly coupled. As increased volumes provide preconditioning for extremes in the entire sea (Pindsoo and Soomere, 2020), higher Baltic Sea volumes generally lead to higher extremes under otherwise similar storm conditions (Weisse and Weidemann, 2017). More specifically, westerly storms not only cause surges on the eastern Baltic Sea coast but also increase the volume of the sea. During subsequent storms, extreme sea levels are then higher than without preconditioning.

Two other noticeable effects may further contribute to Baltic sea level extremes: seiches and meteotsunamis. In the Baltic Sea, seiches with periods of up to tens of hours and e-folding times of up to two days may develop under certain atmospheric conditions (Leppäranta and Myrberg, 2009). Details of these oscillations are still debated and are still not fully understood. From numerical studies, Wübber and Krauss (1979) proposed a series of basin-wide seiches with periods of up to 31 hours. Other authors argued that the existence of such basin-wide oscillations is not entirely

supported by data. They suggested, that such sea level oscillations in the Baltic Sea could alternatively be considered as an ensemble of weakly coupled local seiches with periods between 17 and 27 hours in the Gulf of Riga, the Gulf of Finland, and the Belt Sea (Jönsson et al., 2008). Oscillations with similar periods may also occur in the western Estonian archipelago (Otsmann et al., 2001). When favorably coupled with storm surges or in resonance with atmospheric forcing, such oscillations may contribute to very high sea level extremes at the coast (Suursaar et al.,

2006; Weisse and Weidemann, 2017). Meteotsunamis are generated by moving atmospheric disturbances that trigger resonant sea level fluctuations. While not extremely frequent in the Baltic Sea, cases were described by Pellikka et al. (2014; 2020) for the Gulf of Finland or by Holfort et al. (2016) for the western Baltic Sea. Amplitudes may be in the order of one meter (Pellikka et al., 2014; 2020).

---

2   https://en.ilmatieteenlaitos.fi/wave-height-records-in-the-baltic-sea





The contribution from tides to the height of Baltic sea level extremes is mostly small as the connection to the open
ocean and thus co-oscillation is limited. Overall, tidal ranges are mostly between about 2 and 5 cm (Leppäranta and
Myrberg, 2009). In the western sea areas, tidal ranges of up to 10–30 cm are observed (Leppäranta and Myrberg,
2009). Due to resonance, tidal ranges of up to 20 cm are also found near St. Petersburg at the end of the Gulf of
Finland (Medvedev et al., 2013). While co-oscillation is limited, locally generated tides, albeit small, may also
contribute in the same order of magnitude at some places (Schmager et al., 2008). In the western Baltic Sea, tidal
conditions are predominantly semi-diurnal while in the Gulf of Finland and the Gulf of Riga, diurnal tides prevail
(Medvedev et al., 2013; Schmager et al., 2008). While tidal ranges are generally small, they still have noticeable effects
on the return periods of sea level extremes (Särkkä et al., 2017)

Variability and long-term changes in Baltic extreme sea levels may occur for various reasons but are primarily linked
to changes in relative mean sea level and atmospheric conditions. Relative mean sea level changes will modify the
base upon which other atmospheric drivers of extremes will act although the response is not necessarily linear (e.g.
Arns et al., 2015). For example, for the same wind field, and hence surge levels, higher extremes are expected under
higher relative mean sea levels. Also, changes in the driving atmospheric conditions will lead to changes in the
statistics of waves, surges, etc. which in turn will affect the sea level extremes. Moreover, non-linear interaction
between the contributions, potentially non-stationary behavior of the population of extremes (Kudryavtseva et al.,
2018), and local effects such as from bathymetry or the shape of the coastline may make overall effects on sea level
extremes highly non-additive (e.g., Arns et al., 2015).

Based on tide gauge data and for different periods, several studies revealed positive trends in Baltic sea level
extremes. These trends were found to originate mainly from a corresponding increase in mean sea level (Marcos and
Woodworth, 2018; Ribeiro et al., 2014; Barbosa, 2008) or the increase in the magnitude of the preconditioning
(Soomere and Pindsoo, 2016; Pindsoo and Soomere, 2020). Only for some of the eastern and northernmost stations,
trends showed some contribution from corresponding changes in the large-scale atmospheric circulation and regional
wind patterns (Barbosa, 2008; Ribeiro et al., 2014).

Long-term changes in the wave climate may further contribute to changing extremes through corresponding
adjustments of wave transformation in the surf zone (e.g., wave set-up and swash). So far, there is no conclusive
large-scale figure but results vary strongly depending on period and region. For the Arkona Basin, Soomere et al.
(2012) analyzed wind wave variability and trends based on 20 years of observation and a 45-year wave hindcast.
They concluded that the wave height in this area exhibits no long-term trend but reveals modest inter-annual and
substantial seasonal variations. For shorter periods, estimates from satellite altimetry data, suggest a slight increase in
annual mean significant wave height in the order of 0.005 m yr$^{-1}$. Spatially, wave height increased in the central and
western parts of the sea and decreased in the eastern parts (Kudryavtseva and Soomere, 2017). Nikolkina et al. (2014)
analyzed a multi-ensemble wind wave hindcast covering the entire Baltic Sea using different atmospheric forcing and
periods (1970–2007 and 1957–2008). While these authors found the hindcasts consistently describing the known





spatial patterns with relatively severe wave climate in the eastern parts of the Baltic proper and its sub-basins they could not infer consistent conclusions on long-term changes mainly due to differences in the atmospheric forcing used in the model simulations.

As for mean sea level, extreme sea levels are linked and are correlated with the large-scale atmospheric circulation and its variability. Often, the NAO is used to characterize the state and the variability of the large-scale atmospheric circulation (Johansson, 2014; Marcos and Woodworth, 2018). There are several mechanisms that contribute to the link between Baltic sea level extremes and the NAO. The positive correlation between the phase of the NAO and mean sea level in the northwestern European shelf seas (Woolf et al., 2003) suggests that higher than normal mean sea levels occur during positive phases of the NAO. This would lead to an increase in the baseline sea level upon which wind-induced extremes will act. In addition, there is also a positive relationship between the phase of the NAO and the frequency of westerly winds. Increased frequencies of westerly winds may lead to on average higher-than-normal water volume in the Baltic Sea, which again would increase the baseline. Eventually, potential relationships between changes in the NAO and regional wind patterns would contribute to corresponding changes in wind surges and waves.

Using tide gauge data, Johansson (2014) and Marcos and Woodworth (2018) showed that the positive correlation between the NAO and Baltic sea level extremes persisted even when long-term MSL changes were removed. This indicates that the NAO influences on Baltic sea level extremes are not only limited to the effects of changes in the mean but have contributions from NAO effects on Baltic Sea volume and/or locally generated wind surges and waves. This conclusion was further supported by a model study in which a coupled North and Baltic Sea model was forced solely by wind and sea level pressure from 1948–2011, thereby explicitly excluding effects from global mean sea level rise and rising temperatures (Weidemann, 2014). In this experiment, periods of high water volume in the Baltic Sea occurred more often during positive NAO phases and vice versa, and lower wind speeds were generally needed to sustain higher sea level extremes when the volume was above normal (Weisse and Weidemann, 2017).

Future changes in sea level extremes in the Baltic Sea crucially depend on future changes in relative mean sea level and future developments in large-scale atmospheric conditions and changing wind patterns. For some regions, changes in the frequency or thickness of sea ice may also have an impact. Relative sea level changes will strongly vary across the Baltic Sea because of the existing spatial gradients in GIA. GIA is expected to continue at rates similar to that observed over the last century. Absolute mean sea levels are expected to rise in the entire Baltic Sea, but exact rates are uncertain and depend on models, scenarios, and periods considered (Grinsted, 2015; Hieronymus and Kalén, 2020). Potential future changes in long-term mean and extreme wind speeds are highly uncertain (Räisänen, 2017). Climate model simulations investigated for the Second Assessment of Climate Change for the Baltic Sea Basin (BACC II Author Team, 2015) were highly inconsistent for projected changes in wind speeds at the end of the 21st century (Christensen et al., 2015; BACC II Author Team, 2015). While projections of future wind waves and storm surges in the Baltic Sea are thus highly dependent on the atmospheric scenario, climate model, and



realization used for the projection (e.g. Groll et al., 2017), regions with expected increases in relative mean sea level are highly likely to experience an increase in sea level extremes.

## 2.3 Coastal erosion and sedimentation

### 415    2.3.1 Sources of data

Coastal erosion and sedimentation, and longer-term shoreline changes tend to be researched at local, sediment compartment, or regional scales, where particular combinations of wind, wave, sediment characteristics and availability, and sea-level variations and extremes at a range of scales can be considered. Generally, remote sensing data have been widely applied to investigate coastline change at various spatial and temporal scales. Different sources of remote sensing data include satellite images (e.g., Tiepold and Schuhmacher, 1999), aerial photographs (e.g., Furmanczyk et al., 2011; Dudzinska-Nowak, 2017), air-born laser scanning data (e.g. Hartleib and Bobertz, 2017), and orthophoto maps (e.g., Zhang et al., 2017; Dudzinska-Nowak, 2017). Such data were used, for example, to quantify the rates of coastline change over the past decades along the sandy southern Baltic Sea coast and more recently for the Russian waters (Ryabchuk et al., 2020). Also, historical maps with scales between 1:250000 and 1:5000 dating back to the 1820s have provided useful information on the coastline change of the Baltic Sea at a longer time scale (Deng et al., 2017a; Hartleib and Bobertz, 2017). As tides are small in the Baltic Sea, the baseline of the seaward-most foredune or cliff, which represents the cumulative effect of deposition or erosion, has been used as an indicator to quantify coastline change (e.g., Dudzinska-Nowak, 2017; Zhang et al., 2015).

### 2.3.2 Variability and change of erosion and sedimentation

The Baltic Sea region can be geologically divided into the uplifting Fennoscandian Shield in the North and the subsiding lowlands in the South (details see Fig. 2.8 in Harff et al., 2007). Coastline change is characterized by a corresponding north-south gradient along which the main driving force gradually shifts from GIA to atmospheric and hydrodynamic forcing. Vertical land movement ranges from uplifts of up to almost 9 mm yr$^{-1}$ in the North to subsidence of up to 2 mm yr$^{-1}$ in the South. Since the onset of the Holocene, these vertical movements have caused a persistent marine regression at the northern coasts and a marine transgression along the southern coasts of the Baltic Sea (Harff et al., 2007; 2011). The transitional area between the northern uplift and the southern subsidence is located at the southern coast of the Gulf of Finland (Rosentau et al., 2017).

About half of the shores of the Baltic Sea are comprised of either extremely resistant bedrock or relatively slowly changing cliffs of limestone or morainic materials. Due to this geology and the ongoing GIA, coastal erosion is not regarded as an issue of concern in Finland and for the majority of the Swedish coast (Pranzini and Williams, 2013). The other half of the Baltic Sea shores located primarily in the South and the East are sedimentary and susceptible to coastline change due to erosion and accretion. This part of the coast is characterized by a series of barrier islands and sandy dunes connected with soft moraine cliffs. A typical cross-shore dune profile at the southern Baltic coast





features an established or a series of foredune ridges with typical heights between 3 and 12 m above mean sea level (Łabuz et al., 2018). At the backshore behind the established foredune ridges, drifting or stabilized dunes in transgressive forms are commonly developed. The source of sediment for dune development includes fluvioglacial sands from eroded cliffs, river-discharged sands, and older eroded dunes (Łabuz, 2015).

As with other (semi-) enclosed seas, dominant processes relevant for erosion and accretion on Baltic Sea shores differ from those described in the classic coastal process literature. Differences arise from the lack of substantial tides, the lack of long wind-generated swell, the frequently occurring high angles of storm waves approaching the shore, and partially the presence of a seasonal ice cover. Wave energy and direction are the dominant drivers of sediment transport, erosion, and accretion in the Baltic Sea. Regional properties of the energy supply and the wave-driven transport are described in Soomere and Viška (2014), Kovaleva et al. (2017), or Björkqvist et al. (2018). A broader contextualization for the entire Baltic Sea can be found in Hünicke et al. (2015) and Harff et al. (2017).

Erosion is generally largest during storms when nearshore waves are longest, highest, and thus most energetic. This is primarily the consequence of two effects. First, higher winds will cause higher and, if the duration is sufficient, longer waves. Second, storm surges and wave set-up will increase the nearshore water level so that the water can reach and mobilize sediment at higher locations through swash. As depth is a limiting factor for wave height that may reach the nearshore, an increase in the average water level allows higher waves to come closer to the coast providing higher energy for erosion and sediment transport. The short-term average water level at the shoreline during the time of high waves thus strongly controls the extent and location of erosion. Systematic synchronization of water level and wave intensity may considerably modify the width of the affected nearshore strip (Soomere et al., 2017a).

The classic cut-and-fill concept (e.g. Brenninkmeyer, 1984) assumes that the most energetic steep waves induce beach erosion and mostly cross-shore transport of sediment to the deeper part of the shore while sandbar formation, transport of sediment onshore, and the accretion of the same beach appear during calmer wave conditions with less energy and longer wave periods (constructive swell). Changes on the beach thus primarily follow the incident wave energy level (Masselink and Pattiaratchi, 2001). This cycle is less significant in the Baltic Sea where the wave regime is highly intermittent and contains very small proportions of low-intensity constructive long swell waves (Broman et al., 2006; Soomere et al., 2012). In such conditions, the classic cut-and-fill cycle of beach change is modulated by a relatively intense alongshore movement of sediment.

As tides are generally small in the Baltic Sea, Baltic Sea volume, storm surges, wave set-up, the presence or absence of sea ice, and long-period wave energy from infragravity or edge waves are the main factors influencing the location on the beach profile where sediment can be mobilized and erosion may occur. While strongest alongshore and cross-shore sediment transport in the nearshore (surf and swash zones) usually take place during extreme wave events, the most rapid shoreline changes (both erosion and accretion) occur when high waves attack the shore at relatively large angles, in particular when the angle of wave approach is unusual for the specific location. The latter indicates that specific shore segments may be sensitive to erosion for a particular wave direction only. Small pocket or headland-





confined beaches with very small amounts of sand may thus evolve in a step-like manner (Soomere and Healy, 2011). They remain unchanged and seemingly stable for long periods until a storm from an unusual direction causes massive change. The most extreme erosion events will occur when such a combination comes along with high, normally 480 storm-surge-related, water levels.

Compared to average conditions, in the Baltic Sea, storm waves often approach the shore at relatively large angles (Soomere and Viška, 2014; Pindsoo and Soomere, 2015). On open shores, such waves drive much more intense alongshore transport than waves of comparable height that approach the shore almost perpendicularly. When the approach angle exceeds a threshold of about 45 degrees (Ashton et al., 2001), the predominance of high-angle waves 485 can lead to the explosive development of large spits and sand ridges. The growth of such structures has been observed in the eastern part of the Gulf of Finland (Ryabchuk et al., 2011b; Ryabchuk et al., 2020).

The presence of sea ice during a storm may modify this general pattern (Omstedt and Nyberg, 1991). The hydrodynamic forces are particularly effective in reshaping the shore when no sea ice is present and when the sediment is mobile (Orviku et al., 2003; Ryabchuk et al., 2011a). Storm surges are generally higher in the absence of 490 sea ice. During extreme storm surges, strong waves may reach unprotected and unfrozen mobile sediment on higher sections of the shore that are out of reach for the waves during times with average water levels (Orviku et al., 2003).

Because of the dominant role of westerly winds that take up more than 60% of the year over the southern Baltic Sea (Zhang et al., 2011a) and the sheltering effect of the land in the west, wind waves are typically more developed in the eastern Baltic Sea than in the western part. As a result, sediment transport and dune development are generally more 495 facilitated along the eastern and southeastern coast. For example, the coastal dunes with the largest size and height are developed along the Polish and the Lithuanian coast. On open-ocean facing beaches, periods with wave energy sufficiently high to initiate the transport of substantial amounts of sediment generally occur year-round. Because of the relatively long wave periods and consequently relatively intense wave refraction, waves typically approach the shoreline from small angles but may still cause substantial sediment transport. Being open to large hydrodynamic 500 loads over short periods, but with almost no low and long-period swell that is typically associated with beach accretion, Baltic Sea shorelines are conceptually vulnerable to erosion. Many beaches such as pocket beaches are, however, stabilized by the (mis)match of the directions of predominant strong storms and the geometry of the shoreline; that is, they are only relatively seldom hit by storms and their bay heads are typically geometrically protected. Waves in the Baltic Sea are also relatively short, implying a narrow surf zone and less powerful wave run- 505 up than on the open ocean shores. Consequently, many Baltic Sea beaches with very small amounts of sand are in a fragile but yet almost equilibrium state (Soomere and Healy, 2011).

The Baltic Sea wave climate is highly intermittent and anisotropic following the intermittency and anisotropy of the wind climate; that is, there are only short periods and few directions from which the higher waves approach the shorelines. Only about 1% of the total onshore annual energy flux arrives within the calmest 170–200 days, about 510 60% arrives within 20 days, and as much as about 30% arrives during the 3–4 stormiest days (Soomere and Eelsalu,





2014). A simple consequence of the intermittency and anisotropy of wave fields and the complicated geometry of the coast is that the evolution of Baltic Sea shores is a step-like process (Soomere and Healy, 2011): A few events cause rapid changes when strong waves arrive from specific directions during events with high water levels (Tõnisson et al., 2013b). But for most of the time changes are very slow and require high-resolution measurements such as laser

scanning techniques (Eelsalu et al., 2015; Sergeev et al., 2018) to be detected. The anisotropy of the wave climate combined with the orientation of large sections of the shoreline leads to a large-scale asymmetry of the sediment flux (that is mostly counter-clockwise on the shore of the Baltic Sea proper; Soomere and Viška, 2014; Figure 3). Owing principally to the overall sediment deficit on the Baltic Sea sedimentary shores (Pranzini and Williams, 2013), actual sediment transport is frequently limited by the availability of suitably sized sediment. Thus, the real sediment flux is

only a few percent of the potential flux (Soomere and Viška, 2014). Consequently, there is only slow sediment accumulation in flux convergence areas (Figure 3) and accumulation features may be destroyed easily as sediment transport during a single storm from an unusual direction may be substantially higher. This emphasizes the role of storms and storm waves in the Baltic Sea as substantial factors controlling sediment transport and coastal change.

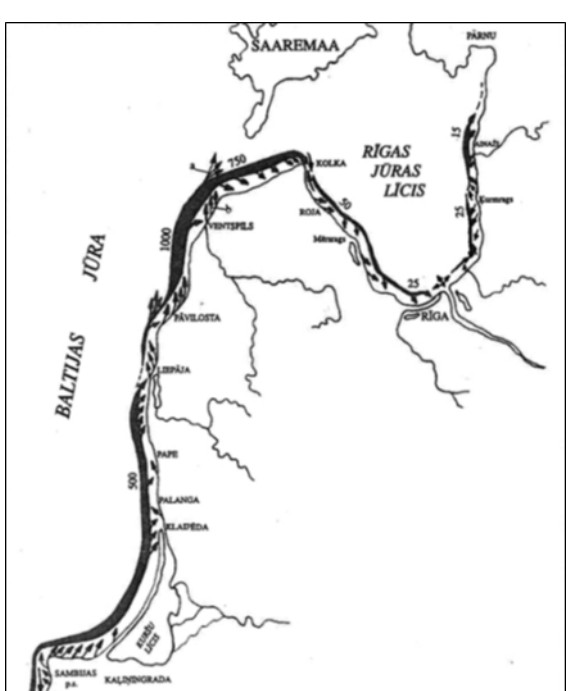
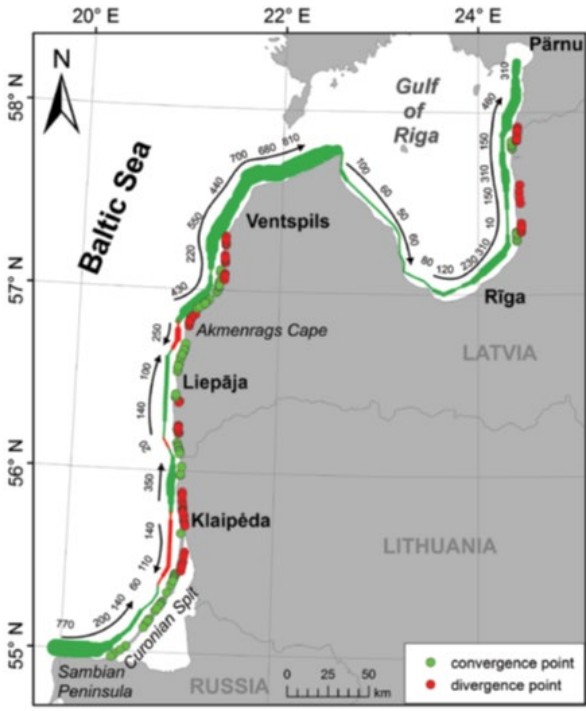

**Figure 3: Direction (arrows) and magnitude (numbers at arrows, in 1000 m³) of net sediment transport: left - original scheme by**
**R. Knaps, amended by Ulsts (1998); right - simulated potential net sediment transport (Viška and Soomere, 2013). Reproduced**
**with permission from the authors and Baltica.**

Hydrodynamic conditions strongly affect coastal morphogenesis. Along the southern and eastern Baltic Sea coast coastal morphogenesis has been extensively studied for more than a century (Keilhack, 1912; Kolp, 1978; Kliewe,





1995; Lampe et al., 2007; Zhang et al., 2010; 2014; Tõnisson et al., 2013a; Furmanczyk and Musielak, 2015; Harff et

al., 2017; Deng et al., 2019). Barrier coasts are generally resilient to changing climate and can maintain their morphology provided that there is a neutral or positive sediment budget (Zhang et al., 2014) and beach migration is unimpeded (Cooper et al., 2020). The foredunes form a natural barrier for coastal protection along a major part of the southern Baltic coast. The part of the Baltic coastline that is protected by engineering structures or newly formed foredune ridges has been able to sustain its general shape and function (both ecological and economical) in the past

decades, while most of the remaining parts, including the soft cliffs and old dune sections, have been subject to continuous and increased erosion (Łabuz, 2015). Most coastline erosion along the southern Baltic Sea is caused either by storms or by human-induced depletion of sediment supply (e.g., as a side effect of engineering structures). Existing studies reveal a highly nonlinear relationship between storm intensity and the rate of coastline erosion along the southern Baltic Sea coast (Zhang et al., 2011b, 2015). Additionally, a strong relationship has been found between

rates of coastline change and the relative level of human development. Even local and modest level of development (e.g., beach nourishment and pier construction) was found to influence the long-term coastline change at larger spatial scales (up to 100 km) (Deng et al., 2014; Dudzinska-Nowak, 2017). Because of this and the strong dependence on the angle at which the waves approach the shore, even smaller human interventions or a climate-related change in the predominant wind and wave directions in the Baltic Sea may substantially alter the structural patterns and pathways

of wave-driven transport and functioning of large sections of the coastline (Viška and Soomere, 2012).

Human activities have become significant drivers of coastal change. With sediment transport being largely confined to shallow water, even smaller coastal construction works such as small boat harbors, can significantly disrupt natural sediment transport pathways, leading to significant local (and frequently undesirable) coastal changes. Ship traffic, with the advent of strongly powered and fast large vessels, has become a significant driver of coastal processes and

has caused erosion locally (Soomere et al., 2009). As is the case worldwide (UNEP, 2019), the demand for coastal sand for construction, industry, and beach nourishment will become increasingly significant for sediment supply.

## 3 Knowledge gaps

Baltic sea level dynamics and coastal erosion have been extensively studied for much longer than a century. Considerable progress has been made on both, understanding details in the Baltic Sea but also enhancing general

knowledge. This was possible due to the outstanding and long data records available in the Baltic Sea and the fact that processes relevant for understanding sea level dynamics and coastal erosion vary across short spatial distances and cover a broad range of time scales (Harff et al., 2017). Even so, when we compiled and reviewed the available publications and knowledge, we identified several gaps. We suggest that addressing these issues may substantially increase our knowledge of the Baltic and more generally contribute to an improved understanding of sea level



dynamics, erosion, and coastline change. Without claiming completeness, and from our perspective, we here provide a list of a few of the more relevant issues.

### 3.1 Long-term changes

A range for possible future Baltic relative mean sea level rise by the end of the 21st century assuming a scenario of strong greenhouse gas emissions (RCP 8.5) was provided by Grinsted (2015). These estimates include contributions

of the thermal expansion of the water volume in the North Atlantic, the transfer of water mass towards the coastal regions, the land movement caused by the GIA, and the contribution from melting of the Antarctic and Greenland ice sheets. In these estimates, the melting of the ice sheets represents the most uncertain factor. These processes are complex to simulate, as they include the dynamics of marine glaciers that are affected not only by the surface heat flux but also by ocean temperature variability and trends at very small spatial scales. Assessments of the contribution

of such processes to sea level rise thus currently rely on estimates based on a mixture of model simulations that are difficult to validate and expert assessments (Bamber et al., 2019). For the Baltic sea level, the largest uncertainty arises from the melting of the Antarctic ice sheet. This is a consequence of contributions from self-gravitational effects, which for the Baltic Sea are affected much stronger from the Antarctic than from Greenland melting (Mitrovica et al., 2001).

For the Baltic Sea coast, as for many coastal areas worldwide, there is a large variety of stakeholders with different levels of risk aversion, who need information on future sea level on different time and spatial scales (Madsen et al., 2019b; Gerkensmeier and Ratter, 2018; González-Riancho et al., 2017). For the Baltic Sea, the potential in providing such information has not yet been fully exploited. There are gaps in providing information covering the whole range of available emission scenarios as well as in contextualizing such information. The latter includes the development of

approaches on how evidence for high-end and plausible upper limits may be accounted for in reliable and acceptable ways (e.g., Thejll et al., 2020). This, in turn, calls for transdisciplinary approaches in which trust and co-creation of knowledge are fostered (Weisse et al., 2019; Schaper et al., 2019) and for the developments of frameworks (e.g., Stammer et al., 2019) accepted by both, the scientific community and a wider range of stakeholders.

Coastal changes are strongly linked to changing extreme sea levels and strong wave conditions. Estimation and

assessment of trends for extremes over centuries are still limited due to observational constraints and the rareness of extremes. Digitalization of the available historical data from analog archives or proxy analyses may improve the situation. Similarly, providing more comprehensive information on potential future developments comprising a larger range of available scenarios would be a step forward. In this context, approaches that account for the non-linear interplay between mean and extreme sea level changes in front of ongoing coastline changes are required. This is

needed in particular, as it is known that the various drivers that in combination with extreme water levels in the Baltic Sea are not well described by classical statistical distributions (Johansson et al., 2001; Männikus et al., 2019). Also,





parameters of extreme value distributions vary in both, space (Soomere et al., 2018) and time (Kudryavtseva et al., 2018).

There is an urgent need for improved modeling of extreme sea levels at finer spatial resolutions that are applicable to
small coastal segments, which are sensitive to minor shifts in wind directions and intensities. This also applies to the
modeling of wave properties as spatial resolutions of wave fields of about 500 m and even finer are necessary to
account adequately for contributions of coastal processes such as wave set-up on total nearshore water levels
(Pindsoo and Soomere, 2015; Pindsoo and Soomere, 2020). Moreover, there are indications that the empirical
distribution of set-up heights in some coastal sections follows an inverse Gaussian distribution that substantially
complicates the construction of joint probability distributions of different components of extreme water levels
(Soomere et al., 2020). There is also a need to better understanding the mechanisms and the long-term changes
(mean, frequency, variability) of seiches and meteotsunamis.

Long-term changes in Baltic mean and extreme sea levels are strongly influenced by GIA. The quantification of land
vertical velocities caused by GIA thus represents an important issue and source of uncertainty in determining long-
term sea level changes. Generally, long-term GIA changes occur jointly with shorter-term land movements caused for
instance by construction works, gas and oil exploitation, or others. Nowadays land vertical velocities are measured
relative to the geoid using the GNSS. However, such GNSS records are still short and show rather patchy patterns of
land vertical velocities (Richter et al., 2012) in which the detection of GIA signals is hampered by short-term
fluctuations. Moreover, trend differences over two decades of available data in vertical positions of GNSS antennae
from stations separated by a few hundreds of kilometers may reach values of up to 2 mm yr$^{-1}$, which is comparable to
the climate signal (Richter et al., 2012). Extrapolating these trends to the end of the 21$^{st}$ century yields relative sea-
level rise uncertainties in the order of 20 cm, which can be substantial for some applications.

More recent publications have attempted to provide gridded estimates of GIA uplift based on a denser network of
local observations  (Vestøl et al., 2019). The uncertainty in these gridded data sets is in the order of 0.25 mm yr$^{-1}$ in
areas close to the local station data but rapidly grows to values between about 1–3 mm yr$^{-1}$ in areas where nearby
station data are unavailable. Along the Baltic Sea coast, the typical uncertainty is of the order of about 0.5 mm yr$^{-1}$,
which is smaller than in previous estimates, but still substantial compared to the climate-related sea level rise. This
calls for a more accurate estimation of the GIA and other contributions to vertical land movements, e.g. by combining
observational data with the results of Earth's crust deformation models. A basin-wide database of historical tide gauge
data with a common reference would strongly support such efforts.

The GIA is a process of adjustment towards a new state of equilibrium in which the Earth's crust viscously adjusts to
changes in ice load. While for the time scales of contemporary sea level rise GIA is usually considered as a constant,
GIA-related vertical velocities will become smaller over millennia and become zero once the Earth's crust has
completely adjusted. Even if such deceleration is small, the question remains, whether it can be detected in sea level
records and if and how it can be separated from the possible climate-related sea level rise acceleration. Except for



Spada et al. (2014), these questions are usually not addressed but may become more pressing in the future when the rates of sea level rise substantially increase relative to the present rates.

## 3.2 Coastal satellite altimetry

Sea level information derived from satellite data has become increasingly important over the past decades. The
estimation of sea level trends and sea level variability from satellite altimetry requires several auxiliary data and environmental corrections to the radar measurements. Besides the internal measurement errors, these corrections account for most of the uncertainties in altimetry. While for most oceanographic applications the correction models are mature for the open ocean, these are still being researched near the coast and in enclosed or semi-enclosed seas. Concerning the Baltic Sea, the most important corrections to be discussed are the wet tropospheric attenuation
correction, the ionospheric corrections, the barometric corrections, and, to a minor extent, ocean tides. The wet troposphere correction is estimated employing radiometers carried on most modern satellites (except ERS-1 and ERS-2). This correction is deteriorated near land, as the radiometer footprint is much larger than the altimeter footprint. As part of the ESA CCI initiative a new composite product was developed (GPD+; Fernandes et al., 2015) which combines different observations (radiometer, GNSS, etc.) in the vicinity of the satellite measurements. The estimation
of ionospheric corrections is based on dual-frequency measurements of the altimeters. Also, this correction fails near the coast, and models, such as Global Ionospheric Maps (GIM), are widely used to replace wrong values near the coast especially in areas with good GNSS coverage. The estimation of the mean sea level often also includes a correction of barotropic pressure change effects. A high-resolution barotropic model (Carrère and Lyard, 2003; Carrere et al., 2016) forced with pressure and wind fields from atmospheric analyses including inverse barometer
corrections is often used. Although the model operates on finite element grids with improved resolution also for the Baltic Sea, the available product is gridded on a 0.25° × 0.25° grid only. Of minor importance, but especially important for variability studies, are ocean tides. Baltic Sea tides are small and often not well assessed in global tide models, which may increase the errors or introduce aliasing effects in variability studies. A recent study (Esselborn et al., 2018) aimed at the importance of the quality of satellite orbit solutions found surprisingly large annual error
signals also for the Baltic Sea when comparing orbit solutions provided by different originators.

Studies of coastal vulnerability, erosion, or the hazard potential along coastlines require sea level estimation as close as possible to the coastline. The conventional radar altimetry is available at a 1 Hz interval (~6.7 km) along-track, and the illuminated footprint varies between 2 and 25 km diameter depending on the surface roughness. Several studies use high-rate data (10 to 40 Hz depending on the altimeter mission), often in combination with advanced
signal processing technologies, such as the so-called re-tracking of individual radar pulses, allowing the extraction of water levels closer to the shore. In recent years in radar altimetry, the SAR/SARin technology has been developed and employed. The first altimeter with SAR/SARin capability is CryoSat-2 (launch 2010) and now is also used for Sentinel-3A and -3B. A study by Dinardo et al. (2018) for the North and Western Baltic Sea using CryoSat-2 data





demonstrated the possibility of mapping the mean sea surface with reasonable quality to as close as 2 km to the coast.
A similar study by Idžanović et al. (2018) for Norway also demonstrated the ability to map the sea surface close to
the coast and under difficult topographic conditions, e.g., in fjords. Both studies also emphasized the importance of
improved environmental corrections replacing standard products.

### 3.3 Coastline changes and erosion

One of the largest gaps is simply the scarcity of in-situ data. Data on coastline changes are mainly collected by
national agencies during geological and coastal surveys and are not systematically available (Madsen et al., 2018).
Such data provide the backbone for estimation of sediment fluxes, the identification of sediment compartments, or for
filling gaps on shoreline relocation and in understanding and forecasting coastline changes. The development of a
systematic database for the entire Baltic Sea would be a major step forward.

Despite intensive research (Pranzini and Williams, 2013), a comprehensive view of alongshore sediment transport
and associated spatial and temporal variability along the subsiding southern Baltic Sea coast is still lacking (Harff et
al., 2017). It is known that the major transport pathway is determined jointly by the angle of incidence of the
prevailing waves and the alongshore currents. Due to the prevailing westerly winds, eastward (counter-clockwise)
transports dominate along major parts of the southern Baltic Sea coast. However, the intensity of secondary transports
induced by easterly and especially by northerly winds is less well understood. In particular along the southern coast,
when storm surges occur with northerly winds the sandy dunes and cliffs are exposed to the highest erosional impact
(Musielak et al., 2017). For some parts of the Baltic Sea coastline, alongshore transport is very sensitive to the angle
of incidence of the waves due to the orientation of the coastline (Viška and Soomere, 2012), and several stretches
have clockwise transport. For example, the incidence angle of westerly wind-waves at the western part of Wolin
Island in Poland (Dudzinska-Nowak, 2017) and the coast of Lithuania and Latvia (Soomere et al., 2017b) is very
small and even small changes in wind direction (in the order of about 10 degrees) could lead to a reverse of the
alongshore transport direction. On these sections of the coast, coastline change is highly variable and extremely
sensitive to future changes in wind-wave climate. Approaches are needed for both better quantification of changes
and uncertainties and frameworks to cope with uncertainties in coastal management.

There are major gaps in the understanding of the functioning of sedimentary compartments and cells and the wave-
driven mobility of sediment between these cells in the eastern Baltic Sea (Soomere and Viška, 2014). Although many
of the cells are quite large, many are also small (Soomere et al., 2007), requiring modeling and measurements at a
much finer scale than is currently available. An approach to comprehensively quantify sediment budget regions with
small-scale sedimentary compartments could be to combine (airborne and terrestrial) laser scanning measurements
(Eelsalu et al., 2015) and detailed bathymetric data with high-resolution, possibly phase-resolving simulations of the
nearshore wave climate. Underwater sediment transport and distribution changes could be roughly estimated from





approximations such as inverse Bruun's Rule (Eelsalu et al., 2015), however more sophisticated techniques should be developed that take systematically into account the alongshore transport.

There is yet to be a universally accepted model of coastal change under sea level rise (Le Cozannet et al., 2019). Indeed, even the assessment of the current state of the world's beaches is incomplete including Europe and the Baltic
Sea (Luijendijk et al., 2018). Simple approximations such as Bruun's rule are continuously applied (e.g. Vousdoukas et al., 2020) despite being widely criticized (Cooper and Pilkey, 2004; Le Cozannet et al., 2016; Cooper et al., 2020). For the Baltic Sea, existing efforts to model shoreline evolution have all focused on relatively short coastal sections (e.g., Zhang et al., 2015). Moreover, they implicitly or explicitly assumed that the wind and wave climate was stationary over the modeled period (e.g., Deng et al., 2015). This assumption has been questioned implicitly by non-
stationary modeling of water level extremes (Kudryavtseva et al., 2018). Extension of modeling capabilities to cover larger coastal segments and to include non-stationary wind and wave conditions is needed. While morphodynamic models are already capable of reproducing the main statistical properties of alongshore sediment transport they are not yet suitable for capturing details of coastal evolution (Deng et al., 2017b). The latter would also include an improved understanding of the factors controlling the formation of the various types of coastal dunes and their
interaction with adjoining morphological features such as cliffs, inlets, and engineering structures or the effect of structures and waterworks on coastline changes.

While there has been substantial progress in the development of coastal morphological models there remains a need for conceptual development (Hinkel et al., 2013; Vitousek et al., 2017) that takes the specifics of Baltic Sea conditions such as the modification of the classic cut-and-fill process or the high sensitivity to minor changes in the
wind and wave climate (Viška and Soomere, 2012) into account. Major progress is expected from research that integrates modeling, measurement, and monitoring (Vitousek et al., 2017).

### 3.4 Decadal predictability of mean and extreme sea levels

Considerable efforts have been made in assessing and understanding past and possible future changes. Assessment of future changes is mostly available for the end of the 21$^{st}$ century or even longer time horizons. They are mostly
available in the form of scenarios or projections, that is in the form of conditional statements depending for example on future greenhouse gas emissions. However, there is a substantial societal need (e.g., Weisse et al., 2009; Weisse et al., 2015) for improved information on the near-term regional climate that has stimulated considerable research in the field of decadal climate prediction (Meehl et al., 2009; 2014). Decadal predictions aim at filling the gap between short-term predictions and long-term projections typically aiming at the years around 2100. It is suggested that there
is some skill required for such predictions obtained mainly from two sources: First, the so-called climate change commitment arises because the warming of the ocean lags that of the land areas so that the atmosphere will continue to warm even if greenhouse gas emissions were stabilized today. Second, the uncertainty from increased greenhouse gas forcing is much smaller for the near future (Meehl et al., 2009).



The techniques required for such predictions have not been, and need to be, explored for Baltic mean and extreme sea
levels. Both are strongly influenced by sea level in the North Atlantic and by the atmospheric and ocean circulation.
The decadal predictability of Baltic sea levels is therefore linked to the predictability of the dynamical state of the
North Atlantic ocean and the large-scale sea level pressure patterns. Decadal predictions are admittedly difficult and
are still in the initial stage of development. However, decadal prediction schemes based on initialized simulations
with Earth System Models display moderate success in the prediction of sea surface temperatures in the North
Atlantic Ocean, which is related partly to the temperature trend caused by climate change and to the genuine
prediction of decadal fluctuations (Müller et al., 2014; Smith et al., 2019). Exploring decadal predictability of Baltic
sea level based on decadal predictions provided by Earth System models possibly downscaled with statistical methods
or with regional models of the Baltic Sea would represent major progress and provide useful information for both, the
scientific community and a large variety of diverse stakeholders.

## 4 Conclusions and key messages

Baltic sea level and coastline change have received considerable attention for centuries. As a consequence, the Baltic
Sea tide-gauge network represents one of the most densely spaced networks comprising some of the longest available
records worldwide. Many of the processes involved in sea level dynamics and coastline change are relevant in the
Baltic Sea where their relevance and contribution vary over short distances and across time scales. This has not only
fostered a regional understanding of the implications of climate change in the Baltic Sea but has also substantially
contributed to extending general concepts and widening our present understanding of sea level and coastline change.
From an ecological perspective, Reusch et al. (2018) considered the Baltic Sea to be a "time machine for the future
coastal ocean" due to the unique combination of an early history of multi-stressor disturbance and ecosystem, the
early implementation of cross-border environmental management, and the availability of long-term data series and
strong scientific foundation providing unique opportunities. We argue that for similar reasons, global sea level
research and research on coastline change may greatly benefit from research undertaken in the Baltic Sea. This is in
particular because of the outstanding data availability and the extremely large number of processes contributing to
observed variability and change in the Baltic Sea. Presently the situation is different. Despite the dense observational
network, the long records available, and the considerable progress made in the general understanding of global
geophysical processes based on Baltic Sea data (Ekman, 2009; Omstedt, 2017), data from the Baltic Sea is often
excluded in global sea level research (Jevrejeva et al., 2006). This is primarily because of the large number of
processes contributing to Baltic Sea level variation and change, which makes interpretation difficult. We suggest that
this represents opportunities rather than threats.



**Author contributions.** The manuscript is a joint effort of all authors. It represents the result from many fruitful discussions
within the BalticEarth community and the review of existing literature. More specifically, the contributions are: AO and RW
drafted the introduction; EZ, BH, TiS, and RW the sections on mean sea level; BH, EZ, KK, TiS, ID, KM, TaS, and RW the
sections on extreme sea level; and TaS, KP and WZ the sections on coastal erosion and sedimentation. The sections on
knowledge gaps, conclusions, and key messages are the result of a joint effort and discussions between all authors. All
authors carefully read, commented, and complemented the entire manuscript. The final manuscript was compiled by RW.

**Competing interests.** The authors declare that they have no conflict of interest.

**Acknowledgments.** This work is a contribution to the Baltic Earth programme (https://baltic.earth) and part of the Baltic
Earth Assessment Reports.

**Financial support.** TaS acknowledges institutional support IUT33-3 by the Estonian Ministry of Education and Research
via Estonian Research Council and the European Economic Area (EEA) Financial Instrument 2014–2021 Baltic Research
Programme project EMP480 "Solutions to current and future problems on natural and constructed shorelines, eastern Baltic
Sea". KP acknowledges the European Regional Development Fund program Mobilitas Pluss, reg. nr 2014-2020.4.01.16-
0024, project MOBTT72. KM received funding from the Danish State through the Danish Climate Atlas.

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
