# Peer review of "Sea Level Dynamics and Coastal Erosion in the Baltic Sea Region"

_Earth System Dynamics, 2021_

## Referee Comment (RC2)

Referee comment on the MS "Sea Level Dynamics and Coastal Erosion in the Baltic Sea Region"
https://doi.org/10.5194/esd-2021-6

The MS is intended to constitute one of the "review papers summarizing and updating the knowledge around the major Baltic Earth science topics. Being part of the series, this study concentrates on sea level dynamics and coastal erosion in the Baltic Sea region." Knowledge on sea level dynamics and coastal erosion has been significantly advanced during the recent 5-10 years and a well readable summarizing paper is highly needed. Besides the scientific importance, the topic is of very high societal significance due to the (not yet well-known) impacts of climate change and the need for appropriate mitigation measures.

The authors are prominent scientists in the field. Extensive list of up-to-date scientific literature is referred. Therefore, the expectations for the MS are high.

Reading the MS further with high interest, I found that different parts are not equally reader-friendly. There is excellent sub-chapter "2.3 Coastal erosion and sedimentation", which is fluent to read, and the sentences contain important findings (including numerical estimates) that are backed by references. Contrary to this, the Abstract and several of the sub-chapters contain occasionally long phrases which information content is not clear. For example, the Abstract begins with: "There are a large number of geophysical processes affecting sea level dynamics and coastal erosion in the Baltic Sea region. These processes operate on a large range of spatial and temporal scales and are observed in many other coastal regions worldwide." I am not able to learn very much out of this very general formulation.  Among "large number of processes", a few most important processes could be outlined. Similar problems are in "4 Conclusions and key messages", which ends with "This is primarily because of the large number of processes contributing to Baltic Sea level variation and change, which makes interpretation difficult."

The readers might be interested to find the conclusive statements easily, instead of going through the extensive list of references. Considering the mean sea level and its trends, including their regional characteristics, inclusion of a figure could be useful, for example, Fig. 9 by Madsen et al. (2019). Regarding absolute SLR, a good candidate could also be https://www.eea.europa.eu/data-and-maps/figures/sea-level-changes-in-europe-october-1992-may-1 (given also at the end of this comment). The sub-chapter on extreme sea levels lacks the numerical estimates about sea level, although there is good description of wind waves that contribute to the run-up in the coastal areas. For clarity, the issues of wind waves could be moved to a separate sub-chapter. Regarding sea level extremes, again some figures could be useful. A candidate could be Fig. 2 by Wolski et al. (2014). Decadal variations of mean and extremes of sea level are missing in the result sections and are considered mainly in the "Gaps of knowledge" section, although publications on identified variations are available and referenced. Baltic Sea Chart Datum 2000 – a common reference level for nautical charts and sea level information in the Baltic Sea – has been or is being established in the riparian countries. Datum corrections of about several tens of cm are introduced. The MS does not reflect this development.

The sub-chapter 3.3 name "Knowledge gaps" is often used in the case when there are policy-driven goals determined beforehand. Such goals are missing for the present review MS; therefore appropriate approach could be to develop something like "further research challenges" or similar. Within this, breakthrough issues could be defined and elaborated, based on new emerging research techniques. It seems that the drivers for new results are remote sensing (including altimetry), modelling at very high resolution, and, perhaps machine learning. I expect the whole sub-chapter 3.3 recomposed, therefore specific remarks are not given.

The MS should be revised in order to make the text more concise and coherent. The Abstract and Conclusions should be elaborated to include distinct findings.

Specific remarks

L40: Baltic Sea area and volume numbers could be referenced by original study, not the overview publication. The newest scientific publication is by Jakobsson et al., 2019.

L42: Perhaps it is not good to start the specific aims with the phrase "may appear irrelevant" but point out what is relevant. The logical structure of the whole sentence is not very clear: "considerable attention over the last centuries" is not related to the global perspective of sea level change as noted in the beginning of the sentence.

L51: "short distances", the scale of distances could be specified.

L52-53: "to study a wide range of phenomena with larger and global relevance" could be specified.

L79: "dynamical consideration", it is interesting to remind that already Svansson (1980) estimated the time scale of Baltic volume change about 12 days, using the analytical model of Helmholz resonator.

L84-85: "About 75% of the basin-average mean sea level change externally enters the Baltic Sea as a mass signal from the adjacent North Sea". It could be clarified what are the processes making the rest 25% of the mean sea level change. The next sentence presents already the sea level variations occurring within the sea (not affecting the mean). From the list of this sentence, mean wind and precipitation affect the mean sea level as well. Freshwater discharge (rivers, P-E) and issues related to ice dynamics are not mentioned in the text. It could be useful to introduce briefly the concept of water budget of semi-enclosed basin. Around Fig. 2, basic principles could be explained in a few sentences. The subsequent paragraph of sediment dynamics is very good and can serve as an example to elaborate the sea level part.

L127-144: The question of different reference levels could be addressed in more detail. The references are rather old. The publication by Bogdanov et al., 2000 (L140) deals specifically with the Kronstadt time series, not with uncertainty of coastal sea level observations in general. Baltic Sea Chart Datum 2000 should be introduced.

L129: The publication by Kowalewska-Kalkowska and Marks, 2011 cannot be found on the web. It could be uploaded to ResearchGate, Academia or some other platform.

L158-160: Introductory sentences could be reformulated to be more informative. For example, I doubt that variations in ocean currents have direct significant effect on Baltic Sea level, compared to the meteorological changes they induce, and to the thermal expansion of seawater.

L165-166: "tide gauge data are more representative for coastal changes", why to say "more", is there a substantial number of historical offshore tidal gauges anywhere?

L187: Nice that comparison has been made, but what are the results?

L196: "Stronger than normal westerly winds push water masses into the Baltic Sea, raising overall sea level". Here, the mechanism of barotropic water exchange through the Danish Straits could be briefly outlined (e.g Stigebrandt, 1983; Mattsson, 1996).

L197-204: The part of the text presenting salinity and its influence should be placed in the logical order. I think the basic driver for salinity is along-basin separation of main saltwater source (Danish Straits) and freshwater sources (largest rivers in the Bothnian Sea and Gulf of Finland), that creates along-basin (estuarine) salinity gradient. Then it could be explained, how this persistent salinity gradient affects sea level via seawater density and what are the mean sea level differences. Further, all the sentences should be checked if there is any result included. For example: "Any changes in the hydrological cycle, either in the long term or in the form of

episodes associated with strongly increased or decreased river runoff or precipitation/evaporation over the Baltic Sea, have the potential to substantially modify this salinity gradient (Kniebusch et al., 2019) and thus the distribution of sea level anomalies and their variability in space and time."

L205-212: From these eight lines, I was able to understand only that several studies have been made on relations between the NAO and Baltic sea level. What are the results?

L212-217: Did I understood correct: Karabil et al. (2018) have shown that atmospheric pressure difference between Bay of Biscay and Northern Norway near Tromsø (termed BANOS index) explains Baltic mean sea level variations better than NAO, due to the proper account of the inverse barometer effect.

L233: "point to slightly higher sea level rise", what is the range?

L237: "Future changes in the Baltic Sea freshwater budget may further affect the regional distribution of Baltic mean sea level". What is the direction of budget change (I guess the full budget is zero due to mass conservation, but the components are changing) and corresponding sea level change? How much?

L266: The sub-chapter "2.2.1 Sources of data" from the "2.2 Extreme sea levels" deals with wave measurements. The title and the text should match each other.

L277-279: In this introductory part, wave set-up has been put on the first place in the extreme sea levels. This view is biased. In the whole sub-chapter, the wave part could be compressed, giving more elaboration to the storm surges, including their occurrence frequency (Wolski and Wiśniewski, 2020).

L284: "preconditioning" should be written without quotation marks, but it needs to be more specified. One type of preconditioning is lower mean sea level before strong westerly winds (Lass and Matthäus, 1996), that favors inflow of highly-saline water (Major Baltic Inflows). We are considering here the basic type of preconditioning, when mean sea level is prior to storm already high (Suursaar et al., 2006; Madsen et al., 2015) and storm surge creates wind sloping and long gravitational waves on top of that.

L288: There is a reference to the publication that is already a review book. I think reviews should refer to the original study results, whenever possible.

L296: For the Gulf of Riga, a reference to the 2005 flood should be added (Suursaar et al., 2006).

L297: The statement that storm surges "are primarily caused by strong onshore winds during storms": the effect of preconditioning from the previous paragraph must be mentioned as well. Important storm surge components are also long gravitational waves, including seiches, and wave set-up.

L337: The height of seiche is more important for the extreme sea level than its period. The heights have been discussed by Wolski and Wiśniewski (2020). When Jönsson et al. (2008) have proposed the seiche period 17 h in the Gulf of Riga, then Suursaar et al. (2002) have presented the period 5 h. This should be mentioned, because the time series plots of sea level frequently reveal such oscillations after sudden changes of wind.

L345-347: The same comment as for L288.

L401-405: Very general wording, key points cannot be found.

L487-492: One effect of sea ice is also land-ocean interaction in the form of shore ride-up and pile-up of ice, see Omstedt et al. (2014) and the references therein.

L552-734: Specific comment on sub-chapter 3.3 were omitted, see the general comments.

L736-741: Is the key message "Research of Baltic sea level and coastal issues is well developed and is societally important".

L742-745: The paper by Reusch et al. (2018) is very interesting, but there is no clue with the present MS so far.

L745-754: The text starting with "We argue that..." belongs rather to the discussion.

Technical remarks

Regarding the style, there is occasionally rather high uncertainty in the text. For example, the word "may" is counted 48 times. (This broke to my eye, therefore I checked some other similar review papers, usually this word is used several times less).

References within this comment

Jakobsson, M., Stranne, C., O'Regan, M., Greenwood, S.L., Gustafsson, B., Humborg, C. and Weidner, E., 2019. Bathymetric properties of the Baltic Sea. *Ocean Science*, *15*(4), pp.905-924.

Lass, H.U. and Matthäus, W., 1996. On temporal wind variations forcing salt water inflows into the Baltic Sea. *Tellus A*, *48*(5), pp.663-671.

Mattsson, J., 1996. Some comments on the barotropic flow through the Danish Straits and the division of the flow between the Belt Sea and the Öresund. Tellus A, 48(3), pp.456-464.

Madsen, K.S., Høyer, J.L., Fu, W. and Donlon, C., 2015. Blending of satellite and tide gauge sea level observations and its assimilation in a storm surge model of the N orth S ea and B altic S ea. *Journal of Geophysical Research: Oceans*, *120*(9), pp.6405-6418.

Omstedt, A., Elken, J., Lehmann, A., Leppäranta, M., Meier, H.E.M., Myrberg, K. and Rutgersson, A., 2014. Progress in physical oceanography of the Baltic Sea during the 2003–2014 period. *Progress in Oceanography*, *128*, pp.139-171.

Stigebrandt, A., 1983. A model for the exchange of water and salt between the Baltic and the Skagerrak. *Journal of Physical oceanography*, *13*(3), pp.411-427.

Suursaar, Ü., Kullas, T., Otsmann, M. and Kõuts, T., 2002. A model for storm surge forecasts in the Eastern Baltic Sea. *WIT Transactions on Modelling and Simulation*, *31*.

Suursaar, U., Kullas, T., Otsmann, M., Saaremae, I., Kuik, J. and Merilain, M., 2006. Cyclone Gudrun in January 2005 and modelling its hydrodynamic consequences in the Estonian coastal waters. *Boreal Environment Research*, *11*(2), p.143.

Svansson, A., 1980. Exchange of water and salt in the Baltic and adjacent seas. *Oceanologica Acta*, *3*(4), pp.431-440.

Wolski, T. and Wiśniewski, B., 2020. Geographical diversity in the occurrence of extreme sea levels on the coasts of the Baltic Sea. *Journal of Sea Research*, *159*, p.101890.

[Figure]

Trend in absolute sea level across Europe based on satellite measurements (1992–2013)

mm/year

−4  −3  −2  −1  0  1  2  3  4  5  6  7

---

## Author Response (AR1)

We thank both reviewers for their constructive comments that have helped us to clarify and improve some points in our manuscript. In the following, we show how we addressed the individual issues raised by the reviewer in the revised manuscript.

**Reviewer #1**

In the first paragraph of the review, the referee states that parts of the manuscript are "too generic and sometimes superficial". While the reviewer acknowledges that "a good number of references" is "included that will point the reader towards detailed results" he/she argues that the "manuscript must be extended to include at last the major results". In the original manuscript, we intentionally preferred general assessments rather than specific numbers that are available from the referenced literature. The latter typically vary depending on, for example, the periods considered, the methods used, etc. However, we acknowledged the point that this might appear too generic. To address the issue we revised the manuscript by including more specific results and numbers from the referenced literature, specifically on the magnitude of sea level variability, its regional distribution, and the values of sea level extremes as measured by tide gauges and derived from models.

In the first bullet point, the referee provided several specific suggestions related to the same point. These were accounted for as follows:

- We included a map illustrating the spatial distribution of mean sea level changes.
- Similarly, a map for extremes was added.
- We further added a figure illustrating the relation between Baltic Sea level and the large-scale atmospheric circulation.
- Also, a map of measured coastline changes for the Baltic Sea was included.

The specific points raised by the referee were addressed as follows (line numbers refer to the original manuscript):

- Lines 84-85: We agree that this is misleading. In the referenced publication, it means that if we refer to the external trend; that is at the entrance of the Baltic Sea (1.6 mm/year) as "100%", then local values may deviate by as much as 25% caused by redistribution within the Baltic Sea. We reformulated this part to make this clear and explicit.
- Line 246: We added the additional reference suggested by the referee.
- Line 255-257: We agree that this was misleading. The acceleration referenced only refers to the observational period. We revised the manuscript and clarified that the contribution is only small if we assume that it does not change in the future. The latter appears unlikely if ice sheets (in particular Antarctica) melt more rapidly in the future.
- Line 294-303: Storm surge heights were included in the discussion.
- Line 336 (and in many other places throughout the manuscript): A map with local names was included.
- Lines 356-357: We did not include this suggestion as it does not represent the only reason.
- Lines 362-363: We agree. The point was not fully/correctly addressed in the original manuscript. Corresponding statements and discussions were included. We also added the suggested reference and removed the one used in the original manuscript because the latter is a summary of the reference suggested by the referee.
- Line 374: The authors of the referenced study tested the "ability" of the classic methods for extracting the trend of 0.5 mm/yr (using linear regression and the Mann-Kendall method) against the presence of normally distributed random error with a standard deviation of 0.5 m. Both methods indicated the presence of the trend. For the actual data, the linear regression method revealed statistically significant increases in basin-wide significant wave heights for 1993-2015 (typically 20,000-40,000 snapshots of wave fields each year) whereas the Mann-

Kendall method did not. The total change was about 12 cm that is about 10% of the overall significant wave height. We clarified this point in the revised manuscript.

- Line 387: Pressure was added.
- Lines 403-407: We partly agree. Strictly and as suggested by the reviewer, the point belongs to the previous section. However, it is needed here for understanding and contextualization, as changes in the mean will affect the extremes. To address the point, we moved it to the previous section but also restated it here and made it clearer that changes in the extremes depend on changes in the mean.
- Lines 454-461: The comment is true, but we think restating these concepts is essential for contextualization of the Baltic Sea specifics. To address the point we rewrote the paragraph with a somewhat stronger focus on the Baltic Sea and clearer links to the following paragraphs.
- Section 3.3: We agree but think that the various types of missing data need to be filled concurrently, as we say at the end of the section. However, we also recognize that the model output of drivers (in the case of the Baltic coast, this is primarily waves) is important, and indeed some efforts are on the way, running wave models at a resolution suggested by the reviewer. We provided some additional sentences to that effect in the revised manuscript. However, it does not negate any of the statements made so far in section 3.3.
- Minor errors and typos were corrected as suggested.

**Reviewer #2**

In the first two paragraphs, the referee acknowledged the extensive list of up-to-date scientific references and the timeliness of this review paper. We thank the reviewer for this assessment.

In the subsequent paragraphs, the reviewer addressed several shortcomings which he/she would like to see addressed in a revised version.

In the third, fourth, and sixth paragraphs, the reviewer addressed the different levels of detail provided in the different sections. Several suggestions are made, how this can be addressed. We took these points into account and carefully re-worked the mentioned parts of the manuscript to make them more concise and coherent. Specifically, we

- inspected and screened the manuscript for long phrases with limited informational content. These were deleted and replaced with shorter and more targeted statements.
- as suggested by the reviewer, explicitly mention examples of processes in the abstract and in section 4.
- highlighted important findings making conclusive statements easier to identify.
- included maps illustrating the spatial distribution of mean and extreme sea level changes (see also reply to Referee #1).
- added to and extended the discussion on decadal variability.

The reviewer further mentioned the Baltic Sea Chart datum as a common reference level for nautical charts and that the manuscript does not reflect this development. We briefly mentioned and acknowledged this development in the revised version although we did not see that this has any impact on the assessment provided and the conclusions drawn.

The reviewer suggested moving the discussion of wind waves into a separate sub-chapter. This suggestion makes sense as wind waves are usually averaged out when sea level is considered. On the other hand, waves do have effects on sea level such as wave set-up and a combined discussion also makes sense from our perspective. For example, wave-driven effects create the largest danger in terms

of flooding during extreme water level events where they contribute up to 1/3 of the local water level increase in some locations. To address the point raised by the reviewer we renamed the chapter into "Extreme sea levels and wind-generated waves".

Regarding section 3.3 the reviewer stated that the phrase "knowledge gaps" is "often used in the case when there are policy-driven goals determined beforehand. Such goals are missing for the present review MS". The reviewer is right that such goals are indeed missing in our manuscript as they are well beyond the scope of the manuscript. We were not aware that this phrase is used in this context only. Our manuscript is part of a series of review papers, the general structure of which was agreed upon among the different groups beforehand. This also included the use of the phrase "knowledge gaps" in the specified context. To balance the trade-off between keeping the agreed structure and considering the reviewer's point we now use "Knowledge gaps and further research challenges" as a sub-chapter title in the revised manuscript.

The reviewer further requested that sub-chapter 3.3 should be recomposed but did not give any specific comment apart from the one discussed above. As provision and discussion of policy goals are outside the scope of this manuscript as well as of the entire series of review papers to which this manuscript belongs we, therefore, refrained from including such goals and from re-composing the entire section. However, we carefully checked the section to conclusively highlight our perspective on future research needs.

The specific points raised by the referee were addressed as follows (line numbers refer to the original manuscript):

- Line 40: We updated the numbers according to the suggestion.
- Line 42: The comment suggests that our statement in the manuscript was misleading. We intended to illustrate that (from a global perspective) the Baltic Sea is small and may appear irrelevant for the global figures. We revised this sentence to avoid this misinterpretation.
- Line 51: We added corresponding numbers.
- Line 52-53: We added examples to illustrate these points.
- Line 79: Thanks for the additional and useful reference. The point raised and the reference were included.
- Line 84-85: We agree that this was misleading. In the referenced publication, it means that if we refer to the external trend; that is, at the entrance of the Baltic Sea (1.6 mm/year) as "100%", then local values may deviate by as much as 25% caused by redistribution within the Baltic Sea. We reformulated this part to make this clear and explicit. We further explicitly and briefly discuss freshwater discharge in the text referring to Figure 2 (now Figure 3) where these processes are present. Sea ice effects are discussed in relation to erosion and sedimentation in section 2.3.2.
- Line 127-144: We briefly introduced the Baltic Sea Chart Datum 2000 (see above) and correct the quotation of Bogdanov et al. (2000). The question of different reference levels is relevant as it hampers comparison and may introduce uncertainties. We think that it should be briefly mentioned but that a comprehensive discussion is beyond the scope of the paper.
- Line 129: Unfortunately we can not upload publications from other authors to the web. We quote this publication as a presentation given at the Scientific Symposium "200 years of tide-gauge measurements in Świnoujście", Świnoujście, Poland, November 18, 2011.
- Lines 158-160: We assume that the reviewer refers to currents within the Baltic Sea. We intended to refer to currents in general. For example, if the Meridional Overturning Circulation in the North Atlantic changes, it will modify the large-scale sea level in the North Atlantic and as a result also in the North and Baltic Seas. We reformulated and clarified the context in which this sentence is meaningful.

- Lines 165-166: The formulation was misleading. The intention was to state that tide gauges are located mostly at the coast. It was reformulated.
- Line 187: We don't understand this comment. The results are explicitly presented in Table 1, which is clearly referred to in the text.
- Line 196: We followed the suggestion and briefly outlined the mechanism.
- Line 197-204: We revised this along the line suggested by the reviewer.
- Line 205-212: An additional sentence was added describing the relation between NAO and Baltic Sea level variability. An important result is, that correlation is not stable but variable across time indicating that NAO is not the optimal pattern and/or sampling effects are important. We made this clear and more explicit in the revised manuscript.
- Line 212-217: Yes, the link between the interannual variations of Baltic winter sea level and the NAO is spatially and temporally heterogeneous (stronger in the north, weaker in the south, stronger in recent years, and weaker in the past). The link to the Biscay-Tromsø difference is temporally and spatially more stable. This pressure difference dominated the NAO in all regions until approximately 1970. Thereafter, it does so only in the south, whereas in the north the impact of both SLP patterns is now comparable. In general terms, the Biscay-Tromsø SLP difference is a better indicator of Baltic sea level interannual variations than the NAO, although the NAO is also a good indicator for the last two decades. We improved the description in the revised manuscript.
- Line 233: Numbers from the study were added.
- Line 237: The reviewer is thinking of the redistribution of water mass within the Baltic Sea, but the changes in the freshwater budget affect sea level through changes in salinity. For example, if in the future the Baltic Sea becomes overall fresher, sea level will rise on average. So in total sea level change is not necessarily zero. Since precipitation is one of the variables whose change is more difficult to predict, the quantification of this effect in the future is complicated.
- Line 266: We modified the titles as suggested above including waves.
- Line 277-279: We rearranged the text and extended the discussion of storm surges as suggested.
- Line 284: Quotation marks were used to make a definition. We now use italic instead. In addition, the reviewer is right and the definition needs to be more specified. We addressed this point as suggested by the reviewer.
- Line 288: We changed the text and refer to the study by Winsor, P., Rodhe, J., Omstedt, A. (2001). Baltic Sea ocean climate: an analysis of 100 yr of hydrographic data with focus on the freshwater budget. Climate Research 18: No. 1-2, pp. 5–15. We also included the recent estimate from Mohrholz (2018). Major Baltic Inflow Statistics – Revised. Front. Mar. Sci. 5, art no 384. doi:10.3389/fmars.2018.00384 stating that "... the top limit of transport through the Danish straits is about 45 $km^3$ $day^{-1}$. This corresponds to a maximum mean sea level change of 12 cm $day^{-1}$"
- Line 296: The reference was added.
- Line 297: To be able to address the processes individually, we prefer not to conflate them but follow the definitions used in Pugh and Woodworth (2014): Sea Level Science, Cambridge Univ. Press (see Chapter 7 and glossary). Here storm surge refers to the changes of sea level in coastal waters caused by winds and air pressures acting on the sea surface. We revised the text and the headlines to make this clear.
- Line 337: We added the suggested reference and extended the discussion as suggested.
- Lines 345-347: We cited the secondary references as the original is not available to us. To make this clear we changed the quotations to (Witting, 1911, as cited in LM 2009) and as (Defant 1961, as cited in LM 2009).

- Lines 401-405: The reviewer is right. The text was revised making it more specific by including numbers and key messages.
- Lines 487-492: Yes, we agree and we added some more sentences on sea ice effects.
- Lines 552-734: See our reply to the general comment on section 3.3 above.
- Comments on Lines 736-741, and 742-745:: We intended to provide the message that the Baltic Sea (i) offers a rich data set and (ii) is affected by many processes, so (iii) it represents a good laboratory to study coastal change. We revised the text to make these points and the quotation of the Reusch et al. (2018) manuscript clearer.
- Lines 745-754: We prefer to retain this as this appears to be mostly a matter of style but we removed the phrase "We argue …".
- Technical remarks: We browsed the manuscript and reduced the use of "may" wherever possible.

**Sea Level Dynamics and Coastal Erosion in the Baltic Sea Region**

Ralf Weisse[1], Inga Dailidiene[2], Birgit Hünicke[1], Kimmo Kahma[3], Kristine Madsen[4], Anders Omstedt[5], Kevin Parnell[6], Tilo Schöne[7], Tarmo Soomere[6], Wenyan Zhang[1], Eduardo Zorita[1]

[1]Institute of Coastal System Analysis and Modeling, Helmholtz Zentrum Hereon, Max-Planck-Str. 1, 21502 Geesthacht, Germany
[2]Klaipeda University, Marine Research Institute, Klaipeda, LT-92294, Lithuania
[3]Finnish Meteorological Institute, Helsinki, Finland
[4]Danish Meteorological Institute, Copenhagen, 2100, Denmark
[5]University of Gothenburg, Department of Marine Sciences, Box 461, SE-405 30 Göteborg, Sweden
[6]Tallinn University of Technology, School of Science, Department of Cybernetics, Tallinn, 12618, Estonia
[7]German Research Centre for Geosciences GFZ, Potsdam, 14473, Germany

*Correspondence to*: Ralf Weisse (ralf.weisse@hereon.de)

**Abstract.** There are a large number of geophysical processes affecting sea level dynamics and coastal erosion in the Baltic Sea region. These processes operate on a large range of spatial and temporal scales and are observed in many other coastal regions worldwide. Together with the outstanding number of long data records, this makes the Baltic Sea a unique laboratory for advancing our knowledge on interactions between processes steering sea level and erosion in a climate change context. Processes contributing to sea level dynamics and coastal erosion in the Baltic Sea include the still ongoing visco-elastic response of the Earth to the last deglaciation, contributions from global and North Atlantic mean sea level changes, or from wind waves affecting erosion and sediment transport along the subsiding southern Baltic Sea coast. Other examples are storm surges, seiches, or meteotsunamis contributing primarily to sea level extremes. All such processes have undergone considerable variations and changes in the past. For example, over the past about 50 years, the Baltic absolute (geocentric) mean sea level rose at a rate slightly larger than the global average. In the northern parts, due to vertical land movements, relative mean sea level decreased. Sea level extremes are strongly linked to variability and changes in  large-scale atmospheric circulation. Patterns and mechanisms contributing to erosion and accretion strongly depend on hydrodynamic conditions and their variability. For large parts of the sedimentary shores of the Baltic Sea, the wave climate and the angle at which the waves approach the nearshore are the dominant factors, and coastline changes are highly sensitive to even small variations in these driving forces. Consequently, processes contributing to Baltic sea level dynamics and coastline change are expected to vary and to change in the future leaving their imprint on future Baltic sea level and coastline change and variability. Because of the large number of contributing processes, their relevance for understanding global figures, and the outstanding data availability,  global sea level research and research on coastline changes  may greatly benefit from research undertaken in the Baltic Sea.

**1 Introduction**

Regional climate change in the Baltic Sea basin has been systematically assessed in two comprehensive assessment reports:
BACC I (BACC Author Team, 2008) and BACC II (BACC II Author Team, 2015) initiated by BALTEX and its successor
Baltic Earth (https://baltic.earth). As a follow-up, the present study represents one of the thematic Baltic Earth Assessment
Reports (BEARs) which consists of a series of review papers summarizing and updating the knowledge around the major
Baltic Earth science topics. Being part of the series, this study concentrates on sea level dynamics and coastal erosion in the
Baltic Sea region.

The Baltic Sea is an intra-continental, semi-enclosed sea in northern Europe that is connected to the Atlantic Ocean only
via the narrow and shallow Danish Straits (see Figure 1 for a map of geographical names). With an area of about
393,000417,000 km$^2$ and a volume of about 21,200 km$^3$ (Jakobsson et al., 2019)(Leppäranta and Myrberg, 2009), it
contributes less than a tenth of a percent to the area and the volume of the global ocean (Eakins and Sharman, 2010).
For that reason, contributions from the Baltic Sea to global sea level changes are small. Over the last centuries and
decades,
While from a global perspective sea level and coastline changes in the Baltic Sea may appear irrelevant, they have
received considerable attention in research anyway. There are over the last centuries and decades for several reasons:

1.  Historically, changes in sea levels and coastlines have influenced Baltic Sea harbors, settlements, and economic
    activity over millennia. As a result, the area comprises not only some of the longest available tide-gauge
    records worldwide, but also much longer observational evidence that challenged our understanding of sea level
    dynamics and land movements associated with the glacial isostatic adjustment (GIA) (e.g., BIFROST project
    members, 1996), and contributed significantly to our present understanding of large-scale sea level changes
    on a global scale.

2.  Processes and forcing contributing to Baltic sea level dynamics and coastline change substantially vary
    substantially over short distances (Harff et al., 2017). For example, there is a pronounced north-south gradient
    in GIA leading to substantial differences in the rates of relative sea level change across the Baltic Sea (see
    below). Together with zonal and meridional changes in the geological composition of the coast and the
    variability of the wind- and wave-driven hydrodynamics this leads to variations in the relative contributions
    of these factors to erosion and sedimentation in the order of kilometers. Time scales of processes and forcings
    also vary considerably, ranging from a few seconds (e.g. wind waves) to millennia (e.g. GIA). Again, this
    enables researchers to study a wide range of phenomena with larger and global relevance (e.g. glacial isostasy
    or characteristics of relative sea level changes).

3.  Finally, and from a regional perspective, regional mean and extreme sea level changes and erosion represent
    important indicators of regional climate variability and change. Any long-term change in mean or extreme sea
    levels as well as in erosion and accretion will have an immediate impact on society, influencing sectors such

as coastal protection, shipping, or development of offshore renewable energy resources among others (e.g., Weisse et al., 2015).

[Figure]

**Figure 1: Map of the Baltic Sea and geographical names used in this manuscript. The Øresund and the Belt Sea are together called the Danish Straits. The Bothnian Bay and the Bothnian Sea together form the Gulf of Bothnia.**

70

Historically, considerable progress in sea level research worldwide was made based on early Baltic Sea observations that indicated that sea levels were falling. Stones carved with runic texts, linking them to the coast, were found quite far away from the present-day coastline. Shallow harbors were gradually abandoned as the water level apparently declined. In the 18$^{th}$ century, Celsius (1743) estimated the rate of falling water levels based on so-called seal rocks (Figure 2, left). Seal rocks were economically important for seal hunting and are therefore well described in the written records (Ekman, 2016).

75

[Figure]

[Figure]

**Figure 12: (Left) The Celsius seal rock at Lövgrunden outside the Swedish city of Gävle on the Bothnian Sea coast (Ekman, 2016). The water level today is about 2 meters below the 1731 mark (photo courtesy of Martin Ekman). (Right) Stockholm annual sea level variations (black) and land rise (red) according to Ekman (2003) [redrawn from Omstedt (2015)].**

80   The reason for the sinking water levels was unclear and debated until it was understood, that in the past thick layers of ice had covered Scandinavia and that sea levels were not falling but instead the land was rising elastically after the ice cover disappeared. The idea of postglacial uplift was proposed in the mid-19th century by Jamieson (1865) and then later by others, although the causes of the uplift were strongly debated. Another major progress in ideas was not possible until new knowledge of the thermal history of the Earth due to changes in Sun-Earth orbital motions was available in

85   the late 19th century and early 20th century (Milanković, 1920).

Figure 21 (right) shows observed Baltic sea level change and land rise for Stockholm, one of the longest available tide-gauge records comprising almost 250 years of data. Apart from the long-term trend, substantial variability on different time scales is inferred. A sketch illustrating processes contributing to such variability is shown in Figure 3. For the Baltic Sea, such processes contributing to such variability can be separated into processes that alter the volume of the

90   Baltic Sea and/or the total amount of water in the basin, and processes that redistribute water within the Baltic Sea (Samuelsson and Stigebrandt, 1996; Svansson, 1980). From analyses of tide-gauge data and dynamical consideration, we know, that processes with characteristic time scales of about half a month or longer can change the volume of the water in the Baltic Sea. Due to the limited transport capacity across the Danish Straits, processes with shorter time scales primarily redistribute water within the Baltic Sea (Johansson, 2014; Soomere et al., 2015; Männikus et al., 2019).

95   At longer time scales, North Atlantic mean sea level changes and effects from large-scale atmospheric variability have the strongest influence on Baltic mean sea level variability and change apart from changes caused by movements in the Earth's crust due to the GIA. For example, using experiments with a hydrodynamic model Gräwe et al. (2019) showed that for the period 1980-2000 aAbout 75% of the basin-averaged mean sea level change externally entereds the Baltic

Sea as a mass signal from the adjacent North Sea while the remainder was caused by above-average zonal wind speeds during that period (Gräwe et al., 2019). Variations in the freshwater budget (precipitation minus evaporation (P-E) and river runoff) can further contribute to seasonal and interannual sea level variations in the Baltic Sea in the order of centimeters to decimetres (Johansson, 2016; Hünicke and Zorita, 2006). Long-term average runoff exceeds P-E by about a factor of ten (Leppäranta and Myrberg, 2009). However, presently no long-term trends in the freshwater budget are observed (Johansson, 2016; Rutgersson et al., 2014) that would add to presently observed rates of Baltic sea level change. On shorter time scales, processes such as changing wind- and atmospheric pressure patterns  are the primary processes redistributing water masses within the Baltic Sea and are responsible for  sea level variations occurring within the Baltic Sea (Hünicke and Zorita, 2006) and its semi-enclosed subbasins (Männikus et al., 2019). Other processes such as seiches or wind waves have effects on the height of the sea surface on very short time scales ranging from seconds to hours.

[Figure]

[Figure]

**Figure 3: Processes contributing to sea level variability and change in the Baltic Sea [ redrawn and modified from Johansson (2014)] **

The primary control of the position of the shoreline is sea level (Harff et al., 2017). Much of the Baltic Sea shoreline (particularly in Finland and Sweden) is rock or consolidated sediments, which typically change over decades to millennia. Shorelines composed of unconsolidated sediments typically erode when sea level rises and accrete when sea level falls. However, several other factors modify the control that sea level exerts. Sediment availability and supply are fundamental. A useful conceptual model is the coastal sediment budget, with sediment transport being considered within sediment compartments, which operate essentially independently, particularly with respect to alongshore sediment transport. Sediment compartments  can be very small (hundreds of meters of shoreline) but also can range up to hundreds of kilometers. Where the sediment budget is in deficit, erosion occurs; where it is in surplus, the shoreline accretes. Overlying any long-term trend are shorter-term changes to the shoreline. Changes that occur after a storm event are typically called erosion, although technically if the sediment remains in the active sediment compartment, this use of the term in this context may be seen as inaccurate. The Baltic Sea differs from many other coastal locations. In particular, short wind fetches mean that swell waves are generally insignificant (Broman et al., 2006; Soomere et al., 2012) and waves often approach the shoreline at a large angle (Soomere and Viška, 2014). That means that the Baltic Sea shorelines are very sensitive to  wind direction. Wave periods are short, leading to the situation where waves can approach the shore at high angles, and the effects of an individual storm on beach processes and sediment transport within a compartment can vary considerably depending on wind (and therefore wave)

characteristics, overlying the effects of short term elevated water levels. Figure 4 provides a sketch of the main topographic coastal features and processes contributing to erosion and sedimentation at the Baltic Sea shores.

[Figure]

135 Figure 4: Main coastal topographic features and erosion/accretion processes at the Baltic Sea shores [redrawn and modified from Harff et al. (2017)].

[revised manuscript text omitted]

**2.1.2 Variability, change, and acceleration of Baltic Sea mean sea level**

Long-term changes of relative mean sea levels in the Baltic Sea are dominated by GIA, global sea level change, and other
regional to local scale components and their interaction. mean sea level are the result of several processesThis includes, for
example,ing thermo- and halosteric effectscontributions, or effects from long-term changes in wind and surface air pressure,
ocean currents in the North Atlantic that affect large-scale sea level, or variations in freshwater input in the Baltic Sea , and
gravitational effects. Effects may arise from contributions outside and inside the Baltic Sea (Figure 32).

Global mean sea level rise since the beginning of the last century was estimated from tide gauge records with rates ranging
between about 1–2 mm yr$^{-1}$ (e.g., Oppenheimer et al., 2019). For the era of continuously operated satellite radar altimeters
beginning in 1991, higher estimates ranging between about 3–4 mm yr$^{-1}$ are reported (e.g., Nerem et al., 2018; Oppenheimer
et al., 2019). These trends cannot directly be compared because of the strong decadal variability inherent in the records (e.g.,
Albrecht et al., 2011; Figure 5, right) and their different spatial representativeness.: While Since the majority of tide gauges is
located at the coast, their data are more representative for of coastal changes., Ssatellite data typically reflect open ocean
changes and variability. Also, inhomogeneous data, comprising a few tide-gauge records in the 19$^{th}$ and early 20$^{th}$ centuries
and satellite data with nearly global coverage in the 21$^{st}$ century, hamper quantification and comparability of trends (Jevrejeva
et al., 2008).

For the Baltic Sea, an increase in absolute mean sea level of about 3.3 mm yr$^{-1}$ was estimated based on available multi-mission
1992–2012 satellite altimetry data (Stramska and Chudziak, 2013). This figure is broadly consistent with the global average
rate over that period. Current (1993-2015/2017) altimetry altimetry-derived Baltic Sea mean sea level trends are still
comparable with the global average, with a need for dedicated altimetry processing in the coastal region (Madsen et al., 2019a;
Table 1). Across, the Baltic Sea, the rates of absolute mean sea level rise vary between about 2-3 mm yr$^{-1}$ in the southwestern
parts and about 5–6 mm yr$^{-1}$ in the northern parts for the period 1995-2019 (Passaro et al., 2021; Figure 5, left).

[Figure]

**Figure 5: (Left) Trends in annual absolute mean sea level 1995-2019 derived from multi-mission satellite altimetry. Trends were provided by Baltic Seal (https://balticseal.eu). Data and methods are described in Passaro et al. (2021). [Redrawn and modified from Passaro et al. (2021)]. (Right) Time series of relative sea level trends in Warnemünde over successive 40-year periods; that is, the trend in 2020 is representative for 1981-2020, the trend in 1980 for 1941-1980, etc. The orange line represents the median, the yellow area the 5-95% range of all 40-year trends. Note that the vertical land movement at Warnemünde is close to zero (0.55 ± 0.59 mm yr⁻¹; estimate provided by https://www.sonel.org for the last about 11 years; accessed: 28. June 2021) [Redrawn and modified from https://sealevel-monitor.eu/].**

Relative mean sea level changes and their secular trends are strongly affected by vertical land movements and vary considerably across the Baltic Sea. Geologically, the Baltic Sea region is divided into the uplifting Fennoscandian Shield in the North and the subsiding lowland parts in the South (Harff et al., 2007). Land uplift rates in the northern parts are in the order of several millimeters per year and are comparable to the climatically induced rates of sea level rise in the 21$^{st}$ century. Relative mean sea level trends in the Baltic Sea show a corresponding north-south gradient, reflecting these crustal deformation rates due to GIA: In the northern parts, relative mean sea level decreases with a maximum rate of about 8.2 mm yr⁻¹ in the Gulf of Bothnia (Hünicke et al., 2015). This corresponds to the area with maximum GIA-induced crustal uplift (Peltier, 2004; Lidberg et al., 2010). In the southern Baltic Sea, relative mean sea level increases at a rate of about 1 mm yr⁻¹ with a gradient in a northeasterly direction (Richter et al., 2012; Groh et al., 2017). These findings are supported by numerous studies analyzing Baltic relative mean sea level trends on a national basis (Suursaar et al., 2006a; Dailidienė et al., 2012; Männikus et al., 2019). Different observation periods and analysis techniques hamper comparison to some extent. A more comprehensive figure on Baltic sea level trends and variability was provided by Madsen et al. (2019a) who took a statistical approach to combine data from century-long tide gauge records with results from hydrodynamic modeling based on atmospheric reanalysis data, and compared the results with those from satellite altimetry records (Table 1). The analysis again emphasizes that the rate of sea level rise strongly depends on the period considered but that present-day rates are higher than previously observed. This is also supported by an analysis of trends over successive 40-year periods at the tide gauge Warnemünde that shows pronounced variability in decadal sea level trends with present trends in the upper range of the observed values (Figure 5, right).

**Table 1: Rates of mean sea level rise for the Baltic Sea and estimated uncertainty (one standard deviation range) based on (Madsen et al., 2019a).**

|  | Mean sea level rise (mm/year) | Uncertainty (mm/year) |
| --- | --- | --- |
| Statistical model, 1900-1999 | 1.3 | 0.3 |
| Statistical model, 1915-2014 | 1.6 | 0.3 |
| Statistical model, 1993-2014 | 3.4 | 0.7 |
| Satellite data (CCI[1]), 1993-2015 | 4.0 | 1.3 |

[1] *Here CCI refers to ESA Sea Level CCI ECV v2.0 (Quartly et al., 2017; Legeais et al., 2018)*

Atmospheric forcing in the form of wind and precipitation  can affect the basin-average sea level through changes in the total volume of water, but also the internal distribution of water volume within the Baltic Sea basin. Winds, and more

235 particularly the strength of the westerly winds, modulate the exchange of water masses with the North Sea (Gräwe et al., 2019). During periods of weak or moderate wind conditions a two-layer system of in- and outflow prevails in the Danish Straits while during periods of strong winds in- or outflow occurs across the entire water column, the strength of which is controlled by the sea level gradient between the Kattegat and the western Baltic Sea (Sayin and Krauss, 1996; Stigebrandt, 1983; Mattsson, 1996). This is due to the alignment of the prevailing wind directions in this region with the geographical orientation of the

240 connecting straits between the Baltic and the North Seas favor such conditions in which. sStronger than normal westerly winds push water masses into the Baltic Sea, raising overall sea level and vice versa.

On average, there is a strong salinity increase from the Northeast to the Southwest of the Baltic Sea. This gradient results from the along-basin separation between the main saltwater (Danish Straits) and freshwater (rivers discharging into the Gulf of Finland and the Bothnian Sea) sources. It leads, through density, to a corresponding sea level gradient which, together with the volume of the freshwater input and the effects of the prevailing westerly winds, gives rise to a corresponding sea level

245 variation of about 35–50 cm across the Baltic Sea (Ekman and Mäkinen, 1996). Kniebusch et al. (2019) showed that there is pronounced decadal variability in the freshwater forcing leading to a corresponding variability in the along-basin salinity gradient which in turn affects the sea level gradient. Over the period 1900-2008, Kniebusch et al. (2019) also detected a trend in the salinity gradient but found the trend to be sensitive to the exact period chosen and small compared to the decadal

250 variability. Likewise, precipitation anomalies affect the distribution of the Baltic mean sea level through corresponding changes in salinity and water density (Hünicke and Zorita, 2006). Presently no robust long-term trends in the freshwater budget are observed (Johansson, 2016; Rutgersson et al., 2014) that would add to presently observed rates of Baltic sea level change. Precipitation anomalies may, through the corresponding changes in salinity and water density, also affect the distribution of the Baltic mean sea level (Hünicke and Zorita, 2006). On average, there is a strong salinity increase from the Northeast to the

255 Southwest of the Baltic Sea. This average salinity gradient, together with the volume of the freshwater input and the effects of the prevailing westerly winds, gives rise to a corresponding sea level variation of about 35–50 cm across the Baltic Sea (Ekman and Mäkinen, 1996). Any changes in the hydrological cycle, either in the long term or in the form of episodes associated with strongly increased or decreased river runoff or precipitation/evaporation over the Baltic Sea, have the potential to substantially modify this salinity gradient (Kniebusch et al., 2019) and thus the distribution of sea level anomalies and their variability in

260 space and time.

The relation between Baltic sea level variability with the state of the large-scale atmospheric circulation has been the subject of numerous studies (Kahma, 1999; Johansson et al., 2001; Lehmann et al., 2002; Dailidiene et al., 2006; Hünicke and Zorita, 2006; Suursaar et al., 2006a; Johansson and Kahma, 2016; Chen and Omstedt, 2005, 2005; Omstedt et al., 2004). Mostly, these studies focused on relations between the NAO[1] and Baltic sea level. Generally, the positive phase of the NAO is associated

265 with enhanced westerly winds over the Baltic Sea area which causes sea level to rise. CGenerally, correlations strongly vary
* * *
[1] The *North Atlantic Oscillation* (NAO) basically describes a meridional pattern in sea level pressure (SLP) with higher than normal SLP around the Azores and lower than normal SLPs over Island and vice versa. Variability of this pattern is physically linked to the intensity of the westerlies in the European region.

by region with higher values in the northern and the eastern parts and smaller values in the southern parts of the Baltic Sea. Moreover, the magnitude of the correlations is not constant but changed over the 20th century, including both, periods with very high and very low values (Figure 6). This suggests that the state of the NAO is a good but, for some periods, not the optimal indicator to describe Baltic sea level variability associated with the atmospheric circulation. Based on that finding

270 Karabil et al. (2018) suggested an alternative pattern based on the pressure differences between the Bay of Biscay and Tromsø. Correlations between this new index and Baltic sea level variability were found to be more stable, suggesting that this index is better suited to describe the parts of Baltic sea level variability associated with the large-scale atmospheric circulation (Figure 6). Similar to the NAO this new index exhibits pronounced interannual and decadal variability over the past about 120 years but does no show a significant long-term change (Karabil et al., 2018).

[Figure]

275 **Figure 6: (Left) Correlation between the NAO/the new index (BANOS) and winter mean sea level at Stoc**Moreover, correlations were found to be rather variable over the 20th century, including periods with both high and very low values. These results initiated a search for atmospheric patterns that might be connected better, and be more stable related to sea level in the Baltic Sea. Karabil et al. (2018) for example, suggested such a pattern in which a gradient in sea level pressure anomalies is oriented from southwest to northeast directions and in which the two centers of action are located over the Baltic Sea and the Bay of Biscay. Compared to the

280 NAO, the correlation of this pattern with the Baltic Sea mean sea level was found to be more stable, suggesting that this pattern is better suited to describe Baltic sea level variability associated with large-scale atmospheric changes. Karabil et al. (2018) suggested that the possible mechanism behind this link is not so much the direct effect of wind on the ocean surface, but is rather associated with the inverse barometer effect. **kholm (STO), Warnemünde (WAR), and Cuxhaven (CUX) together with the correlation between both indices. (Right): Correlation maps for the periods when the correlation between the NAO and sea-level variability were**

285 **minimum (top) and maximum (bottom). The two white points in the bottom panel were used to construct the new BANOS index. Reproduced from Karabil et al. (2018). Distributed under a CC-BY license.**

The contribution of the different mechanisms behind current sea-level trends can be also estimated from the analysis of simulations with regional ocean models driven by observed atmospheric and global mean sea level forcing (Gräwe et al., 2019). This approach has the advantage that the effects of land movement on the Baltic sea level are explicitly neglected so

290 that the contribution from other factors can be disentangled. In the simulation of Gräwe et al. (2019), the Baltic sea level rose at a rate of about 2 mm yr$^{-1}$ over the past 50 years, a rate that is slightly larger than the global average. Most of this sea level rise in the Baltic Sea was caused by a corresponding increase of sea level in the North Atlantic Ocean. Model results along

with data from Latvian waters (Männikus et al., 2020) further suggest a heterogeneous pattern of sea level rise, with larger rates in the northern and smaller rates in the southeastern Baltic Sea that appears to be the result of a poleward shift of atmospheric pressure systems (Gräwe et al., 2019).

For the future, global mean sea level rise is expected to have the largest impact on future Baltic sea level changes (Grinsted, 2015; Hieronymus and Kalén, 2020). It is expected, that most of the future Baltic absolute sea level rise will be strongly linked with corresponding large-scale changes in the North Atlantic and the factors modulating these changes. These factors are mainly the thermal expansion of the water column, contributions from melting of the Antarctic ice -sheet (Grinsted, 2015), and imprints from the variability and change of the Atlantic Meridional Overturning Circulation (Börgel et al., 2018).

For absolute sea level trends, the relative contribution from future melting of the Greenland and Antarctic ice sheets is the major source for spatial variations across the Baltic Sea. Melting of ice from the large ice sheets leads to changes in the Earth's gravitational field, rotation, and crustal deformation the patterns of which are called sea level fingerprints (Plag and Jüttner, 2001). For any given place on Earth, these fingerprints are specific to the location of the ice melt. Baltic sea level is substantially more sensitive to melting from Antarctica than from Greenland. For example, if an ice volume melts from Antarctica that is equivalent to an increase of the global mean sea level of 10 cm, the increase of the absolute mean sea level in the Baltic Sea would be about 11 cm (+10%). If the same volume melts from the Greenland ice sheet, the response of the Baltic Sea level would be smaller and spatially more heterogeneous. Here the values would vary around zero such that the southern Baltic Sea would experience an increase in absolute sea level of about 2 cm (20% of the globally averaged sea level rise from Greenland melting), while the northern Baltic Sea would expect a corresponding sea level fall (Hieronymus and Kalén, 2020).

For the Swedish coast, Hieronymus and Kalén (2020) provided sea level projections accounting for such variations. When compared to earlier estimates, this resulted in lower projections in 2100. However, as more recent estimates of potential future contributions from Antarctic melting increased (Oppenheimer et al., 2019), Hieronymus and Kalén (2020) found their projections still being broadly consistent with earlier estimates, because their more careful treatment of spatial inhomogeneities was balanced by the larger estimates for Antarctic melting. More recent estimates of potential contributions from Antarctic melting point to slightly higher sea level rise in the Baltic Sea when compared to the global average (Hieronymus and Kalén, 2020). However, this is somewhat compensated by lower recent estimates of dynamical sea level rise in the North Atlantic region (Hieronymus and Kalén, 2020).

[revised manuscript text omitted]

**2.2.2 Variability and change of Baltic sea level extremes and wind-generated waves**

Baltic sea level extremes occur over a wide range of spatial and time scales. Contributions may can arise from phenomena acting at small scales, such as wind wave run-ups occurring at a local scales with variations in the range of seconds, up to phenomena such as variations in the volume of the entire Baltic Sea with characterized by formation time scales of up to a few weeks and even longer persistence (Soomere and Pindsoo, 2016). In between these ranges, other phenomena such as storm surges contribute to sea level extremes. They All these phenomena are generated mostly by meteorological and to some extent by astronomical factors (Weisse and Hünicke, 2019). From a climate perspective, this indicates that any relevant change in meteorological forcing may be associated with corresponding changes in Baltic sea level extremes.

The most prominent and most relevant phenomena contributing to sea level extremes in the Baltic Sea are storm surges, wind waves, and a *"preconditioning"* that leads to increased water volumes and thus sea levels in the entire Baltic Sea before the onset of a storm (Suursaar et al., 2006b; Madsen et al., 2015). This Preconditioning is associated with periods of prevailing westerly winds that increase the sea level gradient across the Danish Straits. In turn, the increased sea level gradient leads to higher inflow and higher Baltic Sea water volumes (Samuelsson and Stigebrandt, 1996). While the long-term average barotropic outflow from the Baltic Sea is about 7 km$^3$ day$^{-1}$ or 80.000 m$^3$ s$^{-1}$ (Winsor et al., 2001) Flows transports across the Danish Straits can reach values of up to about 425 km$^3$ day$^{-1}$ in both directions (Leppäranta and Myrberg, 2009), which corresponds to a sea level change of about 126 cm day$^{-1}$ over the entire Baltic Sea (Johansson, 2014) (Mohrholz, 2018). Major inflow events are associated with typical volumes in the order of about 100 km$^3$, corresponding to a Baltic sea level increase of about 24 cm (Matthäus and Franck, 1992). Typically, such variations have time scales of about 10 days and longer (Soomere and Pindsoo, 2016) while atmospheric variability on shorter time scales primarily leads to a redistribution of water masses within the Baltic Sea basin (Kulikov et al., 2015) or between the Baltic Proper and the Gulf of Riga (Männikus et al., 2019).

Storm surges refer to changes in sea level in coastal waters, caused primarily by strong onshore winds during storms and secondarily by the action of spatially varying atmospheric pressure on the sea surface (Pugh and Woodworth, 2014) and by wave set-up in specific locations (Soomere et al., 2013). They represent a substantial threat for the low-lying coastal areas of the Baltic Sea, in particular in the southwestern parts (Wolski et al., 2014), the Gulf of Finland (Suursaar and Sooäär, 2016; Averkiev and Klevannyy, 2010), the Gulf of Riga (Suursaar and Sooäär, 2016; Männikus et al., 2019; Suursaar et al., 2006b), and the Gulf of Bothnia (Averkiev and Klevannyy, 2010). Comparing 50 years of hourly sea level data from gauges across the entire Baltic Sea, Wolski and Wiśniewski (2020) showed that Pärnu in the Gulf of Riga and Kemi in the Gulf of Bothnia are particularly prone to storm surges. In their analysis, these two gauges were characterized by both, the highest average number of surges per year and the highest average number of hours per year

with extreme sea levels. Regarding height, a surge in November 1824 in the Gulf of Finland was responsible for the highest recorded water level in the Baltic Sea (4.21 m above tide gauge zero in St. Petersburg, Wolski and Wiśniewski, 2020). In the western Baltic Sea, a storm in November 1872 caused water levels to exceed 3 m at many gauges (Feuchter et al., 2013; Wolski and Wiśniewski, 2020). For the Gulf of Riga/the western Baltic Sea, storm surge heights around 2 m/1–1.5 m are typical (Wolski and Wiśniewski, 2020).

[revised manuscript text omitted]

Variability and long-term changes in Baltic extreme sea levels  can occur for various reasons but are primarily linked to changes in relative mean sea level and atmospheric conditions. Relative mean sea level changes will modify the base upon which other atmospheric drivers of extremes will act although the response is not necessarily linear (e.g. Arns et al., 2015). For example, for the same wind field, and hence surge levels, higher extremes are expected under higher relative mean sea levels. Also, changes in the driving atmospheric conditions will lead to changes in the statistics of waves, surges, etc. which in turn will affect the sea level extremes. Moreover, non-linear interaction between the contributions, potentially non-stationary behavior of the population of extremes (Kudryavtseva et al., 2018), and local effects such as from bathymetry or the shape of the coastline  can make overall effects on sea level extremes highly non-additive (e.g., Arns et al., 2015).

Based on tide gauge data and for different periods, several studies revealed  trends in Baltic sea level extremes. These trends were found to originate mainly from a corresponding  change in mean sea level (Marcos and Woodworth, 2017; Ribeiro et al., 2014; Barbosa, 2008) or the increase in the magnitude of the preconditioning (Soomere and Pindsoo, 2016; Pindsoo and Soomere, 2020). For the periods from 1960 onwards Marcos and Woodworth (2017) showed that the height and duration of sea level extremes increased/decreased at sites with increasing/decreasing relative mean sea level and that these trends mostly disappear when the mean sea level signal is removed. Only for some of the eastern and northernmost stations, trends  indicate some contribution from corresponding changes in the large-scale atmospheric circulation and regional wind patterns (Barbosa, 2008; Ribeiro et al., 2014). Using data from a hindcast simulation ignoring GIA, Pindsoo and Soomere (2020) found increases in the height of Baltic sea level extremes for the period 1961-2004/2005. These increases were strongest in the Gulf of Finland, the Gulf of Riga and, the eastern Baltic proper (Figure 7). Pindsoo and Soomere (2020) concluded that the increases along the Swedish coast and the Gulf of Bothnia were almost entirely a result of global mean sea level rise and an increase in the maximum water volume of the entire sea. On the other hand, they noted substantial contributions from stronger local storm surges to explain the higher values along the eastern shorelines.

[Figure]

**Figure 7: Trends of storm-season (from July to June of the subsequent year) maximum water level derived from simulated total water level in 1961–2004/2005. Reproduced from Pindsoo and Soomere (2020) with permission from Elsevier and the authors.**

Long-term changes in the wave climate may further contribute to changing extremes through corresponding adjustments
of wave transformation in the surf zone (e.g., wave set-up and swash). So far, there is no conclusive large-scale figure
but results vary strongly depending on period and region. For the Arkona Basin, Soomere et al. (2012) analyzed wind
wave variability and trends based on 20 years of observation and a 45-year wave hindcast. They concluded that the
wave height in this area exhibits no long-term trend but reveals modest inter-annual and substantial seasonal variations.
For shorter periods (1993-2015), estimates from satellite altimetry data, suggest a slight increase in annual mean
significant wave height in the order of 10% (about 12 cm or 0.005 m yr⁻¹). Spatially, wave height increased in the
central and western parts of the sea and decreased in the eastern parts (Kudryavtseva and Soomere, 2017). Nikolkina
et al. (2014) analyzed a multi-ensemble wind wave hindcast covering the entire Baltic Sea using different atmospheric
forcing and periods (1970–2007 and 1957–2008). While these authors found the hindcasts consistently describing the
known spatial patterns with relatively severe wave climate in the eastern parts of the Baltic proper and its sub-basins
they could not infer consistent conclusions on long-term changes mainly due to differences in the atmospheric forcing
used in the model simulations.

As for mean sea level, extreme sea levels are linked and are correlated with the large-scale atmospheric circulation and
its variability. Often, the NAO is used to characterize the state and the variability of the large-scale atmospheric
circulation (Johansson, 2014; Marcos and Woodworth, 2018). There are several mechanisms that contribute to the link
between Baltic sea level extremes and the NAO. The positive correlation between the phase of the NAO and mean sea

level in the northwestern European shelf seas (Woolf et al., 2003) suggests that higher than normal mean sea levels occur during positive phases of the NAO. This would lead to an increase in the baseline sea level upon which wind- and pressure-induced extremes will act. In addition, there is also a positive relationship between the phase of the NAO and the frequency of westerly winds. Increased frequencies of westerly winds may lead to on average higher-than-normal water volume in the Baltic Sea, which again would increase the baseline. Eventually, potential relationships between changes in the NAO and regional wind patterns would contribute to corresponding changes in wind- surges and waves.

Using tide gauge data, Johansson (2014) and Marcos and Woodworth (2018) showed that the positive correlation between the NAO and Baltic sea level extremes persisted even when long-term mean sea levelMSL changes were removed. This indicates that the NAO influences on Baltic sea level extremes are not only limited to the effects of changes in the mean but have contributions from NAO effects on Baltic Sea volume and/or locally generated wind surges and waves. This conclusion was further supported by a model study in which a coupled North and Baltic Sea model was forced solely by wind and sea level pressure from 1948–2011, thereby explicitly excluding effects from global mean sea level rise and rising temperatures (Weidemann, 2014). In this experiment, periods of high water volume in the Baltic Sea occurred more often during positive NAO phases and vice versa, and lower wind speeds were generally needed to sustain higher sea level extremes when the volume was above normal (Weisse and Weidemann, 2017).

Future changes in sea level extremes in the Baltic Sea crucially depend on two factors: future changes in relative mean sea level and future developments in large-scale atmospheric conditions and associated with changing wind patterns. For some regions, changes in the frequency or thickness of sea ice may also have an impact. Relative sea level changes will strongly vary across the Baltic Sea because of the existing spatial gradients in GIA and the spatial inhomogeneity associated with the uncertain relative contributions of melting from Antarctica and Greenland (see section 2.1.2). For the Baltic Sea, changing mean sea levels are expected to have larger effects on future extremes than changing atmospheric circulation (Gräwe and Burchard, 2012). GIA is expected to continue at rates similar to that observed over the last century. Absolute mean sea levels are expected to rise in the entire Baltic Sea, but exact rates are uncertain and depend on models, scenarios, and periods considered (Grinsted, 2015; Hieronymus and Kalén, 2020, see also discussion in section 2.1.2).

Potential future changes in long-term mean and extreme wind speeds are highly uncertain (Räisänen, 2017). In the fifth assessment report of the Intergovernmental Panel on Climate Change (IPCC), a poleward shift of Northern Hemisphere mid-latitude storm tracks and an ensemble average increase in the NAO index are reported as possible future developments (Kirtman et al., 2013). Because of the large internal variability together with the large variability among models and scenarios, there was, however, only medium confidence in these projected changes (Kirtman et al., 2013). Recent studies indicate, that the poleward shift of the westerlies is most pronounced in summer and less obvious in other seasons (Zappa and Shepherd, 2017). Because storm surges in the Baltic Sea are primarily a winter and fall phenomenon, this implies that contributions from changes in atmospheric circulation to future Baltic Sea storm surge

535 climate remain uncertain. This is in agreement with climate  model simulations investigated for the Second Assessment of Climate Change for the Baltic Sea Basin (BACC II Author Team, 2015) that were highly inconsistent for projected changes in wind speeds at the end of the 21$^{st}$ century (Christensen et al., 2015; BACC II Author Team, 2015).

Despite these uncertainties, some storm surge and wind wave projections seem to report relatively robust changes
540 towards 2100. Using data from a multi-model ensemble, driven with atmospheric data from eight different climate models and the two emission scenarios RCP4.5 and RCP8.5, Vousdoukas et al. (2016) projected an increase in storm surge heights in the order of about 5-10% across the entire Baltic Sea towards the end of the twenty-first century. For wind waves, Groll et al. (2017) reported an increase of about 5-10% in mean and extreme wave heights using data from two realizations of two emission scenarios. On the other hand, using a substantially larger ensemble consisting of data
545 from six climate models and three emission scenarios Dreier et al. (2021) found no consistent long-term changes in wave climate along the German Baltic Sea coast, and their projected changes in extreme wave height varied between about -10% and +6%.  This again suggests that projections of future wind waves and storm surges in the Baltic Sea are  still highly dependent on the atmospheric scenario, climate model, and realization used for the projection.  Nevertheless, because of the larger effect of changing mean sea levels on the extremes, regions
550 with expected increases in relative mean sea level are still highly likely to experience an increase in sea level extremes.

**2.3 Coastal erosion and sedimentation**

**2.3.1 Sources of data**

[revised manuscript text omitted]

On most shorelines, erosion is concentrated over relatively short periods around high tide, but asAs tides are generally very small in the Baltic Sea, elevated average water levels, which provide conditions conducive to erosion, can be

maintained for extended periods (Johansson and Kahma, 2016; Soomere and Pindsoo, 2016). Therefore, Baltic Sea volume, storm surges, wave set-up, the presence or absence of sea ice, and long-period wave energy from infragravity or edge waves are the main factors influencing the duration and location on the beach profile where sediment can be mobilized and erosion may occur. Systematic synchronization of water level and wave intensity may considerably modify the width of the affected nearshore strip and the depth to which profile changes may occur (Soomere et al., 2017a).

The classic cut-and-fill concept of beach erosion and recovery (e.g. Brenninkmeyer, 1984) assumes that the most energetic steep waves induce beach erosion and mostly cross-shore transport of sediment to the deeper part of the shore while sandbar formation, transport of sediment onshore, and the accretion of the same beach appear during calmer wave conditions with less energy and longer wave periods (constructive swell). Changes on the beach thus primarily follow the incident wave energy level (Masselink and Pattiaratchi, 2001). This cycle is less significant in the Baltic Sea where the wave regime is highly intermittent and contains very small proportions of low-intensity constructive long swell waves (Broman et al., 2006; Soomere et al., 2012). In such conditions If waves approach the nearshore at large angles with respect to shore-normal, the classic cut-and-fill cycle of beach change is modulated by a relatively intense alongshore movement of sediment when compared to open ocean shores with a similar wave intensity (Soomere and Viška, 2014).

While strongest alongshore and cross-shore sediment transport in the nearshore (surf and swash zones) usually take place during extreme wave events, the most rapid shoreline changes (both erosion and accretion) occur when high waves attack the shore at relatively large angles (Ashton et al., 2001), in particular when the angle of wave approach is unusual for the specific location. The latter indicates that specific shore segments may be sensitive to erosion for a particular wave direction only. Several Ssmall pocket or headland-confined beaches with very small amounts of sand are in a fragile, but yet almost equilibrium, state as they are geometrically protected against winds from many directions (Caliskan and Valle-Levinson, 2008). may thus evolve in a step-like manner (Soomere and Healy, 2011). They remain unchanged and seemingly stable for long periods until a storm from an unusual direction causes massive change. The most extreme erosion events will occur when such a combination comes along with high, normally storm-surge-related, water levels.

Compared to average conditions in the world oceans, storm waves in the Baltic Sea, storm waves often approach the shore at relatively large angles (Soomere and Viška, 2014; Pindsoo and Soomere, 2015). On open shores, such waves drive much more intense alongshore transport than waves of comparable height that approach the shore almost perpendicularly. When the approach angle exceeds a threshold of about 45 degrees (Ashton et al., 2001), the predominance of high-angle waves can lead to the explosive development of large spits and sand ridges. The growth of such structures has been observed in the eastern part of the Gulf of Finland (Ryabchuk et al., 2011b; Ryabchuk et al., 2020).

The presence of sea ice during a storm  can modify this general pattern (Omstedt and Nyberg, 1991). The hydrodynamic forces are particularly effective in reshaping the shore when no sea ice is present and when the sediment is mobile (Orviku et al., 2003; Ryabchuk et al., 2011a). Storm surges are generally higher in the absence of sea ice (Omstedt and Nyberg, 1991). During extreme storm surges, strong waves may reach unprotected and unfrozen mobile sediment on higher sections of the shore that are out of reach for the waves during times with average water levels (Orviku et al., 2003). Land–ice interaction in the coastal zone is also crucial during ice winters. The principal erosion mechanisms are the wind-driven shore ride-up and pile-up of ice, and ice growth down to the sea bottom with resulting transport of bottom sediment when the ice drifts out to sea (Girjatowicz, 2004; Leppäranta, 2013; Orviku et al., 2011). The former mechanism is also a risk to structures close to the shoreline.

[revised manuscript text omitted]

There remains a need for further high-resolution modeling of the basic process drivers of coastal change, particularly waves and co-functioning of severe waves and high water levels. Wave models used for the Baltic Sea usually have spatial resolutions of about 2–3 nautical miles (see Björkqvist et al., 2018 for an overview). The most recent Baltic Sea wide simulations have resolutions of about 1 nautical mile (Björkqvist et al., 2018; Nilsson et al., 2019). For some confined areas such as Tallinn Bay (Soomere et al., 2007) or the sea area off Helsinki (Björkqvist et al., 2017) modeled datasets are available at higher resolution in the order of a few hundred meters. Given the complex shape of the sedimentary shoreline of the Baltic Sea, a resolution in the order of 500 m for relatively straight sections of the shoreline and up to 100–200 m for more complicated sections is highly desirable to resolve the existing pattern of sedimentary compartments and cells and to better understand the processes behind their development and stability.

[revised manuscript text omitted]

Celsius, A.: Anmärkning om vattnets förminskande så I Östersiön som Vesterhafvet, Kongl. Swenska Wetenskaps
    Academiens Handlingar, 33–50, 1743.

Chen, D. and Omstedt, A.: Climate-induced variability of sea level in Stockholm: Influence of air temperature and
    atmospheric circulation, Adv. Atmos. Sci., 22, 655–664, doi:10.1007/BF02918709, 2005.

Christensen, O. B., Kjellström, E., and Zorita, E.: Projected Change—Atmosphere, in: Second Assessment of Climate
    Change for the Baltic Sea Basin, Team, T. B. A., II (Ed.), Regional Climate Studies, Springer International Publishing,
    Cham, 217–233, 2015.

Cieślikiewicz, W. and Paplińska-Swerpel, B.: A 44-year hindcast of wind wave fields over the Baltic Sea, Coastal
    Engineering, 55, 894–905, doi:10.1016/j.coastaleng.2008.02.017, 2008.

Cooper, J. A. G., Masselink, G., Coco, G., Short, A. D., Castelle, B., Rogers, K., Anthony, E., Green, A. N., Kelley, J. T.,
    Pilkey, O. H., and Jackson, D. W. T.: Sandy beaches can survive sea-level rise, Nature Climate change, 10, 993–995,
    doi:10.1038/s41558-020-00934-2, 2020.

Cooper, J. A. G. and Pilkey, O. H.: Sea-level rise and shoreline retreat: Time to abandon the Bruun Rule, Global and
    Planetary Change, 43, 157–171, doi:10.1016/j.gloplacha.2004.07.001, 2004.

Dailidiene, I., Davuliene, L., Tilickis, B., Stankevicius, A., and Myrberg, K.: Sea level variability at the Lithuanian coast of
    the Baltic Sea, Boreal Environment Research, 11, 109–121, 2006.

Dailidienė, I., Davulienė, L., Kelpšaitė, L., and Razinkovas, A.: Analysis of the Climate Change in Lithuanian Coastal Areas
    of the Baltic Sea, Journal of Coastal Research, 282, 557–569, doi:10.2112/JCOASTRES-D-10-00077.1, 2012.

Dangendorf, S., Hay, C., Calafat, F. M., Marcos, M., Piecuch, C. G., Berk, K., and Jensen, J.: Persistent acceleration in
    global sea-level rise since the 1960s, Nature Climate change, 9, 705–710, doi:10.1038/s41558-019-0531-8, 2019.

Dean, R. G. and Bender, C. J.: Static wave setup with emphasis on damping effects by vegetation and bottom friction,
    Coastal Engineering, 53, 149–156, doi:10.1016/j.coastaleng.2005.10.005, 2006.

Defant, A.: Physical Oceanography, Pergamon Press, New York, NY, 729 pp., 1961.

Deng, J., Harff, J., Giza, A., Hartleib, J., Dudzinska-Nowak, J., Bobertz, B., Furmanczyk, K., and Zölitz, R.: Reconstruction
    of coastline changes by the comparisons of historical maps at the Pomeranian Bay, southern Baltic Sea, in: Coastline
    Changes of the Baltic Sea from South to East, Harff, J., Furmańczyk, K., and von Storch, H. (Eds.), 19, Springer
    International Publishing, Cham, 271–287, 2017a.

Deng, J., Harff, J., Schimanke, S., and Meier, H. E. M.: A method for assessing the coastline recession due to the sea level
    rise by assuming stationary wind-wave climate, Oceanological and Hydrobiological Studies, 44, 7649, doi:10.1515/ohs-
    2015-0035, 2015.

Deng, J., Woodroffe, C. D., Rogers, K., and Harff, J.: Morphogenetic modelling of coastal and estuarine evolution, Earth-Science Reviews, 171, 254–271, doi:10.1016/j.earscirev.2017.05.011, 2017b.

1025 Deng, J., Wu, J., Zhang, W., Dudzinska-Nowak, J., and Harff, J.: Characterising the relaxation distance of nearshore submarine morphology: A southern Baltic Sea case study, Geomorphology, 327, 365–376, doi:10.1016/j.geomorph.2018.11.018, 2019.

Deng, J., Zhang, W., Harff, J., Schneider, R., Dudzinska-Nowak, J., Terefenko, P., Giza, A., and Furmanczyk, K.: A numerical approach for approximating the historical morphology of wave-dominated coasts—A case study of the

1030 Pomeranian Bight, southern Baltic Sea, Geomorphology, 204, 425–443, doi:10.1016/j.geomorph.2013.08.023, 2014.

Dinardo, S., Fenoglio-Marc, L., Buchhaupt, C., Becker, M., Scharroo, R., Joana Fernandes, M., and Benveniste, J.: Coastal SAR and PLRM altimetry in German Bight and West Baltic Sea, Advances in Space Research, 62, 1371–1404, doi:10.1016/j.asr.2017.12.018, 2018.

Dreier, N., Nehlsen, E., Fröhle, P., Rechid, D., Bouwer, L., and Pfeifer, S.: Future Changes in Wave Conditions at the

1035 German Baltic Sea Coast Based on a Hybrid Approach Using an Ensemble of Regional Climate Change Projections, Water, 13, 167, doi:10.3390/w13020167, 2021.

Dudzinska-Nowak, P.: Morphodynamic processes of the Swina Gate coasta zone development (Southern Baltic Sea), in: Coastline Changes of the Baltic Sea from South to East, Harff, J., Furmańczyk, K., and von Storch, H. (Eds.), 19, Springer International Publishing, Cham, 219–255, 2017.

1040 Eakins, B. W. and Sharman, G. F.: Volumes of the World's Oceans from ETOPO1: https://www.ngdc.noaa.gov/mgg/global/etopo1_ocean_volumes.html, last access: 18 April 2018.

Eelsalu, M., Soomere, T., and Julge, K.: Quantification of changes in the beach volume by the application of an inverse of the Bruun Rule and laser scanning technology, Proc. Estonian Acad. Sci., 64, 240, doi:10.3176/proc.2015.3.06, 2015.

Eelsalu, M., Soomere, T., Pindsoo, K., and Lagemaa, P.: Ensemble approach for projections of return periods of extreme

1045 water levels in Estonian waters, Continental Shelf Research, 91, 201–210, doi:10.1016/j.csr.2014.09.012, 2014.

Ekman, M.: The changing level of the Baltic Sea during 300 years: A clue to understanding the earth, Summer Institute for Historical Geophysics, Godby, 155 pp., 2009.

Ekman, M.: The Man behind "Degrees Celsius": A Pioneer in Investigating the Earth and its Changes, Åland Islands, 159 pp., 2016.

1050 Ekman, M. and Mäkinen, J.: Mean sea surface topography in the Baltic Sea and its transition area to the North Sea: A geodetic solution and comparisons with oceanographic models, J. Geophys. Res., 101, 11993–11999, doi:10.1029/96JC00318, 1996.

Esselborn, S., Rudenko, S., and Schöne, T.: Orbit-related sea level errors for TOPEX altimetry at seasonal to decadal timescales, Ocean Sci., 14, 205–223, doi:10.5194/os-14-205-2018, 2018.

1055 Fernandes, M. J., Lázaro, C., Ablain, M., and Pires, N.: Improved wet path delays for all ESA and reference altimetric missions, Remote Sensing of Environment, 169, 50–74, doi:10.1016/j.rse.2015.07.023, 2015.

Feuchter, D., C. Jörg, G. Rosenhagen, R. Auchmann, O. Martius, and S. Brönnimann: The 1872 Baltic Sea storm surge., in: Weather extremes during the past 140 years., S. Brönnimann, and O. Martius (Eds.), Geographica Bernensia, G89, 91–98, 2013.

1060   Furmanczyk, K. and Musielak, S.: Polish spits and barriers, in: Sand and gravel spits, Randazzo, G., Jackson, D. W. T., and Cooper, J. A. G. (Eds.), Coastal research library, 12, Springer, Cham, 181–194, 2015.

Furmanczyk, K. K., Dudzinska-Nowak, J., Furmanczyk, K. A., Paplinska-Swerpel, B., and Brzezowska, N.: Dune erosion as a result of the significant storms at the western Polish coast (Dziwnow Spit example), Journal of Coastal Research, 64, 756–759, 2011.

1065   Gerkensmeier, B. and Ratter, B. M.W.: Governing coastal risks as a social process—Facilitating integrative risk management by enhanced multi-stakeholder collaboration, Environmental Science & Policy, 80, 144–151, doi:10.1016/j.envsci.2017.11.011, 2018.

Girjatowicz, J. P.: Ice thrusts and piles on the shores of the southern Baltic Sea coast (Poland) lagoons, Baltic Coastal Zone, 8, 5–22, 2004.

1070   González-Riancho, P., Gerkensmeier, B., and Ratter, B. M.W.: Storm surge resilience and the Sendai Framework: Risk perception, intention to prepare and enhanced collaboration along the German North Sea coast, Ocean & Coastal Management, 141, 118–131, doi:10.1016/j.ocecoaman.2017.03.006, 2017.

Gräwe, U. and Burchard, H.: Storm surges in the Western Baltic Sea: The present and a possible future, Clim Dyn, 39, 165–183, doi:10.1007/s00382-011-1185-z, 2012.

[revised manuscript text omitted]

        Suursaar, Ü., Kullas, T., Otsmann, M., and Kõuts, T.: A model for storm surge forecasts in the Eastern Baltic Sea, in: Risk
        analysis III: [papers presented at the Third International Conference on Computer Simulation in Risk Analysis and
        Hazard Mitigation (RISK/2002) held in Sintra, Portugal in June 2002], Brebbia, C. A. (Ed.), /WIT transactions on
        modelling and simulation], 31, WIT Press, Southampton, 509–519, 2002.

1405    Suursaar, Ü., Kullas, T., Otsmann, M., Saaremäe, I., Kuik, J., and Merilain, M.: Cyclone Gudrun in January 2005 and
        modelling its hydrodynamic consequences in Estonian coastal waters, Boreal Environment Research, 11, 143–159,
        2006b.

        Suursaar, Ü. and Sooäär, J.: Decadal variations in mean and extreme sea level values along the Estonian coast of the Baltic
        Sea, Tellus A: Dynamic Meteorology and Oceanography, 59, 249–260, doi:10.1111/j.1600-0870.2006.00220.x, 2016.

1410    Svansson, A.: Exchange of water and salt in the Baltic and adjacent seas, Oceanologica Acta, 3, 431–440, 1980.

[revised manuscript text omitted]

Zhang, W., Schneider, R., Harff, J., Hünicke, B., and Fröhle, P.: Modelling of Medium-Term (Decadal) Coastal Foredune Morphodynamics- Historical Hindcast and Future Scenarios of the Świna Gate Barrier Coast (Southern Baltic Sea),, in:

Coastline Changes of the Baltic Sea from South to East, Harff, J., Furmańczyk, K., and von Storch, H. (Eds.), 19,
1490    Springer International Publishing, Cham, 112–140, 2017.

Zhang, W., Schneider, R., Kolb, J., Teichmann, T., Dudzinska-Nowak, J., Harff, J., and Hanebuth, T. J.J.: Land–sea
interaction and morphogenesis of coastal foredunes — A modeling case study from the southern Baltic Sea coast,
Coastal Engineering, 99, 148–166, doi:10.1016/j.coastaleng.2015.03.005, 2015.

Zhang, Z.-h. and Leppäranta, M.: Modeling the influence of ice on sea level variations in the Baltic Sea, Geophysica, 31, 31–
1495    45, 1995.